# The novel anti-CRISPR AcrIIA22 relieves DNA torsion in target plasmids and impairs SpyCas9 activity

Kevin J. Forsberg [1,2¤]*, Danica T. Schmidtke [1], Rachel Werther [1], Ruben V. Uribe [3], Deanna Hausman [1], Morten O. A. Sommer [3], Barry L. Stoddard [1], Brett K. Kaiser [1,4], Harmit S. Malik [1,2]

1 Division of Basic Sciences, Fred Hutchinson Cancer Research Center, Seattle, Washington, United States of America, 2 Howard Hughes Medical Institute, Fred Hutchinson Cancer Research Center, Seattle, Washington, United States of America, 3 Novo Nordisk Foundation Center for Biosustainability, Technical University of Denmark, Lyngby, Denmark, 4 Department of Biology, Seattle University, Seattle, Washington, United States of America

¤ Current address: Department of Microbiology, University of Texas Southwestern Medical Center, Dallas, Texas, United States of America
* kevin.forsberg@utsouthwestern.edu

**Data Availability Statement:** With the exception of structural data, all relevant primary data are within the paper and its Supporting information files. All

## Abstract

To overcome CRISPR-Cas defense systems, many phages and mobile genetic elements (MGEs) encode CRISPR-Cas inhibitors called anti-CRISPRs (Acrs). Nearly all characterized Acrs directly bind Cas proteins to inactivate CRISPR immunity. Here, using functional metagenomic selection, we describe AcrIIA22, an unconventional Acr found in hypervariable genomic regions of clostridial bacteria and their prophages from human gut microbiomes. AcrIIA22 does not bind strongly to SpyCas9 but nonetheless potently inhibits its activity against plasmids. To gain insight into its mechanism, we obtained an X-ray crystal structure of AcrIIA22, which revealed homology to PC4-like nucleic acid–binding proteins. Based on mutational analyses and functional assays, we deduced that *acrIIA22* encodes a DNA nickase that relieves torsional stress in supercoiled plasmids. This may render them less susceptible to SpyCas9, which uses free energy from negative supercoils to form stable R-loops. Modifying DNA topology may provide an additional route to CRISPR-Cas resistance in phages and MGEs.

## Introduction

CRISPR-Cas systems in bacteria and archaea confer sequence-specific immunity against invading phages and other mobile genetic elements (MGEs) [1,2]. In response, MGEs can circumvent CRISPR-Cas systems by evading CRISPR immunity. In its simplest form, evasion requires only a single mutation within a CRISPR target site, which allows a phage or MGE to escape immune recognition [3]. However, CRISPR-Cas systems routinely acquire new spacer sequences corresponding to new sites within phage and MGE genomes [1]. This means that any single-site evasion strategy is likely to be short-lived. Thus, phages also employ forms of

supporting structural data are available from the PDB database under accession number 7JTA.

**Funding:** This work was supported by Howard Hughes Medical Institute (HHMI) to HSM; G. Harold and Leila Y. Mathers Foundation (Mathers Foundation) to HSM; Helen Hay Whitney Foundation (HHWF) to KJF; HHS | NIH | National Institute of General Medical Sciences (NIGMS) to BLS /R01GM105691/; DOE | Office of Science (SC) to /BLS DE-AC02-06CH11357/; Fred Hutchinson Cancer Research Center (The Hutch) to BLS; Seattle University to BKK /Summer Faculty Fellowship/ and Novo Nordisk Foundation (Grant No. NNF20CC0035580) to MOAS. The funders had no role in study design, data collection and analysis, decision to publish, or preparation of the manuscript.

**Competing interests:** The authors have declared that no competing interests exist.

**Abbreviations:** Acr, anti-CRISPR; CFU, colony-forming unit; crRNA, CRISPR RNA; dsDNA, double-stranded DNA; GTDB, genome taxonomy database; Kan$^R$, kanamycin resistance; LB, lysogeny broth; MGE, mobile genetic element; ORF, open reading frame; RM, restriction–modification; SEC, size exclusion chromatography; SGB, species genome bin; sgRNA, single-guide RNA; SpyCas9, *Streptococcus pyogenes* Cas9; SSB, single-stranded binding.

CRISPR-Cas evasion that are less easily subverted. For instance, some jumbophages assemble a proteinaceous, nucleus-like compartment around their genomes upon infection, allowing them to overcome diverse bacterial defenses, including CRISPR-Cas and restriction–modification (RM) systems [4,5]. Similarly, other phages decorate their DNA genomes with diverse chemical modifications such as the glucosylated cytosines used by phage T4 of *Escherichia coli* [6], which can prevent Cas nucleases from binding their target sequence.

MGEs may also overcome CRISPR-Cas systems by inactivating, rather than evading, CRISPR immunity. MGEs encode diverse CRISPR-Cas inhibitors called anti-CRISPRs (Acrs), which allow them to overcome CRISPR-Cas systems and infect otherwise immune hosts [7]. Most known Acrs bind Cas proteins and inhibit Cas activity by restricting access to target DNA, preventing necessary conformational changes, or inactivating critical CRISPR-Cas components [8,9]. The direct inactivation of Cas proteins by Acrs has proven an effective and widespread strategy for overcoming CRISPR immunity [10].

Recent genetic, bioinformatic, and metagenomic strategies have identified many Acrs that independently target the same CRISPR-Cas system [7–10]. Yet, most CRISPR-Cas systems are not inhibited by known Acrs [10]. Thus, many undiscovered strategies to inhibit or evade CRISPR-Cas systems likely exist in nature. Indeed, over half of the genes in an average phage genome have no known function [11]. To uncover new counterimmune strategies, we recently devised a high-throughput functional metagenomic selection to find genes that protect a target plasmid from *Streptococcus pyogenes* Cas9 (SpyCas9), the variant used most frequently for genome editing [12]. Our selection strategy was designed to reveal any gene capable of overcoming SpyCas9 activity in this system, regardless of mechanism. With this approach, we previously described a new phage inhibitor of SpyCas9, called AcrIIA11, which exhibits broad-spectrum anti-Cas9 activity and is prevalent across human gut microbiomes [12].

Here, we describe *acrIIA22*, which was the second most common Acr candidate recovered from our original functional selection. *AcrIIA22* encodes a 54 amino acid protein that impairs SpyCas9 activity. We observe that homologs of *acrIIA22* are found in hypervariable loci in phage and bacterial genomes. Unlike most other Acrs, AcrIIA22 does not bind strongly to SpyCas9 in vitro. Instead, guided by an X-ray crystal structure of AcrIIA22, coupled with mutational and biochemical analyses, we show that AcrIIA22 encodes a DNA nickase. By nicking a supercoiled plasmid substrate and relieving its torsional stress, AcrIIA22 renders the target less susceptible to SpyCas9 activity. AcrIIA22 thus represents a novel mechanism of SpyCas9 evasion, which capitalizes on SpyCas9's preference for negative supercoils to efficiently form R-loops and cleave DNA [13–16]. Such a resistance mechanism could be accessible to diverse MGEs, providing a route to CRISPR-Cas tolerance in many genetic contexts.

## Results

### Functional selection reveals a novel anti-CRISPR protein, AcrIIA22

We recently carried out a functional selection for SpyCas9 antagonism, recovering clones from metagenomic libraries that could potently inhibit SpyCas9 [12]. In this 2-plasmid setup, we used an arabinose-inducible SpyCas9 on an expression plasmid to cleave the *kanamycin resistance* (*Kan$^R$*) gene of a second "target" plasmid. We then grew cultures in SpyCas9-inducing conditions and measured the proportion of colony-forming units (CFUs) that remained kanamycin resistant (Fig 1A). This proportion is a measure of how many clones retained their target plasmid and, thus, how effectively that plasmid withstood SpyCas9 attack. In our previously published work, we describe AcrIIA11, a novel Acr from a metagenomic clone named F01A_2 (Genbank ID MK637582.1), which was the most abundant clone after functional selection of a human fecal microbiome [12]. This functional selection also revealed another

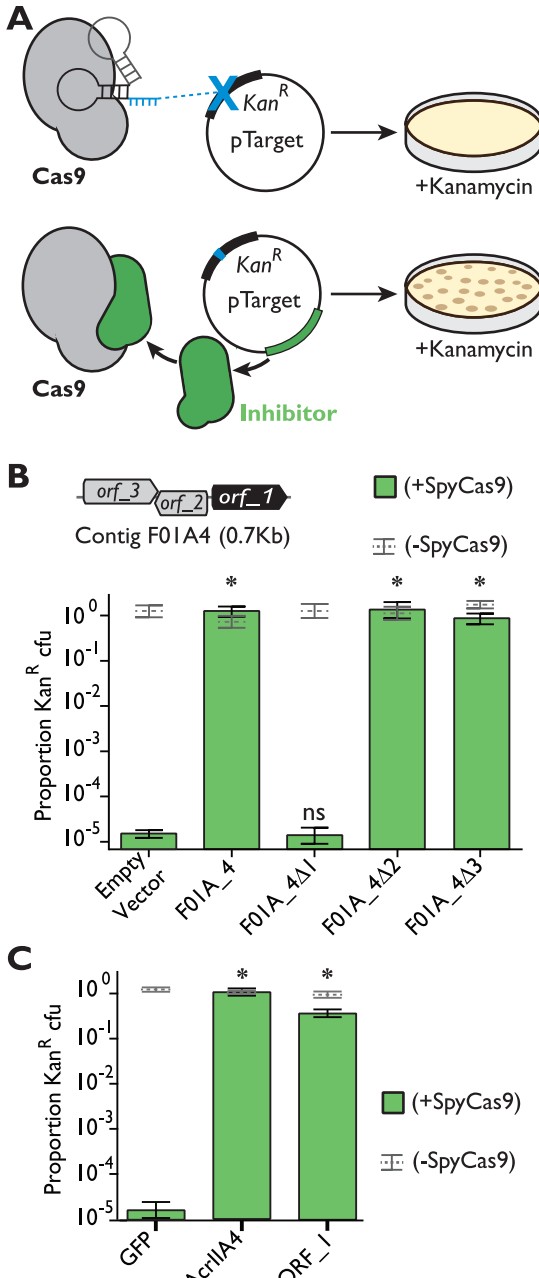

**Fig 1. Functional selection reveals a metagenomic contig encoding a novel SpyCas9 inhibitor.** (**A**) A plasmid protection assay was used to reveal SpyCas9 inhibition. In this assay, plasmids without SpyCas9 inhibitors are cleaved by Cas9 and do not give rise to Kan$^R$ colonies, whereas those encoding inhibitors withstand SpyCas9 attack and yield Kan$^R$ colonies. (**B**) The contig F01A_4 protects a plasmid from SpyCas9 attack, but an early stop codon in *orf_1 (Δ1)* eliminates this phenotype. Stop codons in *orf_2* or *orf_3 (Δ2* and *Δ3)* have no effect. Thus, we conclude that *orf_1* is necessary for inhibition of SpyCas9. Asterisks depict statistically significant differences in plasmid retention between the indicated genotype and an empty vector control in SpyCas9-inducing conditions (Student *t* test, $p < 0.002$, $n = 3$); ns indicates no significance. All *p*-values were corrected for multiple hypotheses using Bonferroni method. (**C**) Expression of *orf_1* (which we name *acrIIA22*) is sufficient for SpyCas9 antagonism, protecting a plasmid as effectively as *acrIIA4*. Asterisks are as in panel B but relate to the GFP negative control rather than to an empty vector. The individual numerical values that underlie the summary data in this figure may be found in S1 Data. Kan$^R$, kanamycin resistance; SpyCas9, *Streptococcus pyogenes* Cas9.

protective clone, F01A_4 (Genbank ID MK637587.1), which was the second most abundant contig following selection. Together, these 2 contigs (F01A_2 and F01A_4) accounted for >96% of the normalized read coverage.

The F01A_4 contig is 685 bp long, encodes 3 potential open reading frames (ORFs), and confers complete protection against SpyCas9, with plasmid retention equaling that of an uninduced SpyCas9 control (Fig 1B). To determine the genetic basis for SpyCas9 antagonism in this contig, we introduced an early stop codon into each of the 3 potential ORFs and analyzed how these mutations affected the contig's ability to protect a target plasmid from SpyCas9. We found that an early stop codon in *orf_1* reduced the proportion of *Kan*[R] CFUs by a factor of $10^5$, matching the value observed for an empty vector control (Fig 1B). Furthermore, expression of *orf_1* alone was sufficient for SpyCas9 antagonism (Fig 1C), protecting a target plasmid from SpyCas9 cleavage as effectively as the potent SpyCas9 inhibitor, AcrIIA4. In this assay, *orf_1* was slightly toxic when singly expressed in *E. coli*, reducing growth rate by 7% (S1 Fig). Combined, our results indicate that *orf_1* completely accounts for the SpyCas9 protection phenotype of contig F01A_4.

One mechanism through which *orf_1* could apparently antagonize SpyCas9 in our functional assay would be by lowering its expression. To address this possibility, we carried out 2 experiments. First, we swapped the *spycas9* gene for *gfp* in our expression vector and asked whether *orf_1* induction impacted fluorescence output. We saw no change in fluorescence upon *orf_1* induction, indicating that *orf_1* neither suppressed transcription from our expression vector nor altered its copy number (S2 Fig). Second, we used western blots to test whether *orf_1* expression impacted SpyCas9 protein levels through the course of a plasmid protection assay. We used a CRISPR RNA (crRNA) that did not target our plasmid backbone to ensure that *orf_1* expression remained high to maximize its potential impact on SpyCas9 expression levels. We observed that *orf_1* expression had no meaningful effect on SpyCas9 expression at any time point (S2 Fig). Thus, we conclude that *orf_1* does not impact SpyCas9's translation or degradation rate. Therefore, *orf_1* must act via an alternative mechanism to inhibit SpyCas9 activity. Based on these findings, we conclude that *orf_1* encodes a bona fide Acr protein and hereafter refer to it as *acrIIA22*.

Next, we investigated whether *acrIIA22* could also allow phages to escape from SpyCas9 immunity (S3 Fig). We measured SpyCas9's ability to protect *E. coli* from infection by phage Mu, in the presence or absence of *acrIIA22*. As a control, we carried out similar phage infections in the presence or absence of the well-established SpyCas9 inhibitor, *acrIIA4*. As anticipated, SpyCas9 significantly impaired Mu when targeted to the phage's genome but not if a nontargeting crRNA was used. Consistent with previous findings [12], phage Mu could infect targeting strains equally well as nontargeting strains when *acrIIA4* was expressed, indicating that SpyCas9 immunity was completely abolished by this *acr*. In comparison, *acrIIA22* improved the infectivity of phage Mu by a factor of 100 to 1,000 across multiple experimental conditions (S3 Fig). We therefore conclude that *acrIIA22* only partially protects phage Mu from SpyCas9 whereas it completely protects plasmids against SpyCas9 cleavage.

## AcrIIA22 homologs are found in hypervariable regions of bacterial and prophage genomes

AcrIIA22 is 54 amino acids in length and has no sequence homology to any protein of known function, including all previously described Acrs. We examined the distribution of *acrIIA22* homologs in NCBI's NR and WGS databases but found just 7 hits, limiting our ability to make evolutionary inferences about its origins or prevalence. We therefore expanded our search to include IMG/VR, a curated database of cultured and uncultured DNA viruses [17], and

assembly data from a meta-analysis of 9,428 diverse human microbiome samples [18]. With an additional 23 unique homologs from these databases, we found that the majority of *acrIIA22* homologs exist in either of 2 genomic contexts: prophage genomes (Fig 2A and S4A Fig) or small, hypervariable regions of bacterial genomes, which we refer to hereafter as "genomic islands" (Fig 2B and S4B Fig). The original metagenomic DNA fragment from our selection, F01A_4, shared perfect nucleotide identity with one of these genomic islands (Fig 2B).

Because most *acr*s are found in phage genomes, we first examined the prophages that encoded AcrIIA22 homologs. These prophages were clearly related, based on many homologous genes and a similar genome organization (S4A Fig). We found that these prophages had inserted into several different bacterial loci, including one site between the bacterial genes *purF* and *radC* (locus #3, S4A Fig). This insertion site is nearly identical to the highly conserved sequences that flanked *acrIIA22*-encoding bacterial genomic islands (S4B Fig). Based on this common insertion site, we hypothesize that the apparently bacterial genomic islands with *acrIIA22* homologs originated from a common prophage insertion at this locus. We speculate that the original *acrIIA22*-encoding bacterial genomic island resulted from the incomplete excision of an ancestral, *acrIIA22*-encoding prophage. Supporting this hypothesis, *acrIIA22* homologs are always found at the end of prophage genomes, near their junction with a host bacterial genome (Fig 2A and S4A Fig).

To better understand *acrIIA22*'s gene neighborhood, we again searched the assemblies of over 9,400 human microbiomes for more examples of these genomic islands [18]. We did not include *acrIIA22* as a query. Instead, we only considered contigs with ≥98% nucleotide identity to *purF* and *radC*, the conserved genes that flanked the genomic islands. This search yielded 258 contigs. Aligning these sequences revealed that each contig encoded a short, hypervariable region of small ORFs, which was flanked by conserved genomic sequences (Fig 2B and S4B Fig). In total, we observed 128 unique examples of these hypervariable loci, which displayed considerable gene turnover, resulting in 54 distinct gene arrangements among the 128 unique loci. Despite not being included in our search criteria, *acrIIA22* homologs were universally conserved in all 128 unique genomic islands. In contrast, no other gene was present in more than two-thirds of the 54 distinct gene arrangements (Fig 2B and S4C Fig). Based on this finding, we infer that the arrival of *acrIIA22* preceded the diversification seen at this locus and has been retained despite the considerable gene turnover that has occurred subsequently.

Though most ORFs in these islands were of unknown function, many had close homologs in the genomes of 9 representative *acrIIA22*-encoding phage (S4A–S4C Fig). This suggests that phages continue to supply the genetic diversity seen at these hypervariable genomic loci. These rapid gene gains and losses likely occur as they do in other genomic islands, via recombination between this locus and related MGEs that infect the same host bacterium, without the MGE necessarily integrating into the locus [19]. Taken together, our data suggest that an incomplete prophage excision event left *acrIIA22* behind in a bacterial genomic locus, which then diversified via gene exchange with additional phage genomes (Fig 2C and S4D Fig).

Like in the genomic islands (Fig 2B), we found *acrIIA22* homologs in hypervariable regions of prophage genomes, where they were consistently near the junction with a host bacterial genome (S4A Fig). Thus, nearly all *acrIIA22*-encoding loci show signatures of frequent recombination. Despite this, we could find no gene consistently present within or outside of *acrIIA22*-encoding genomic islands that could account for their hypervariable nature (e.g., an integrase, transposase, recombinase, or similar function that is typically associated with locus-specific recombination [20]). Instead, *acrIIA22* was the only gene conserved at this locus. This conservation led us to speculate that *acrIIA22* might promote recombination, either alone or with other factors. If this were true, it could explain the high rates of gene exchange observed adjacent to the *acrIIA22* gene in phage and bacterial genomes (Fig 2 and S4 Fig).

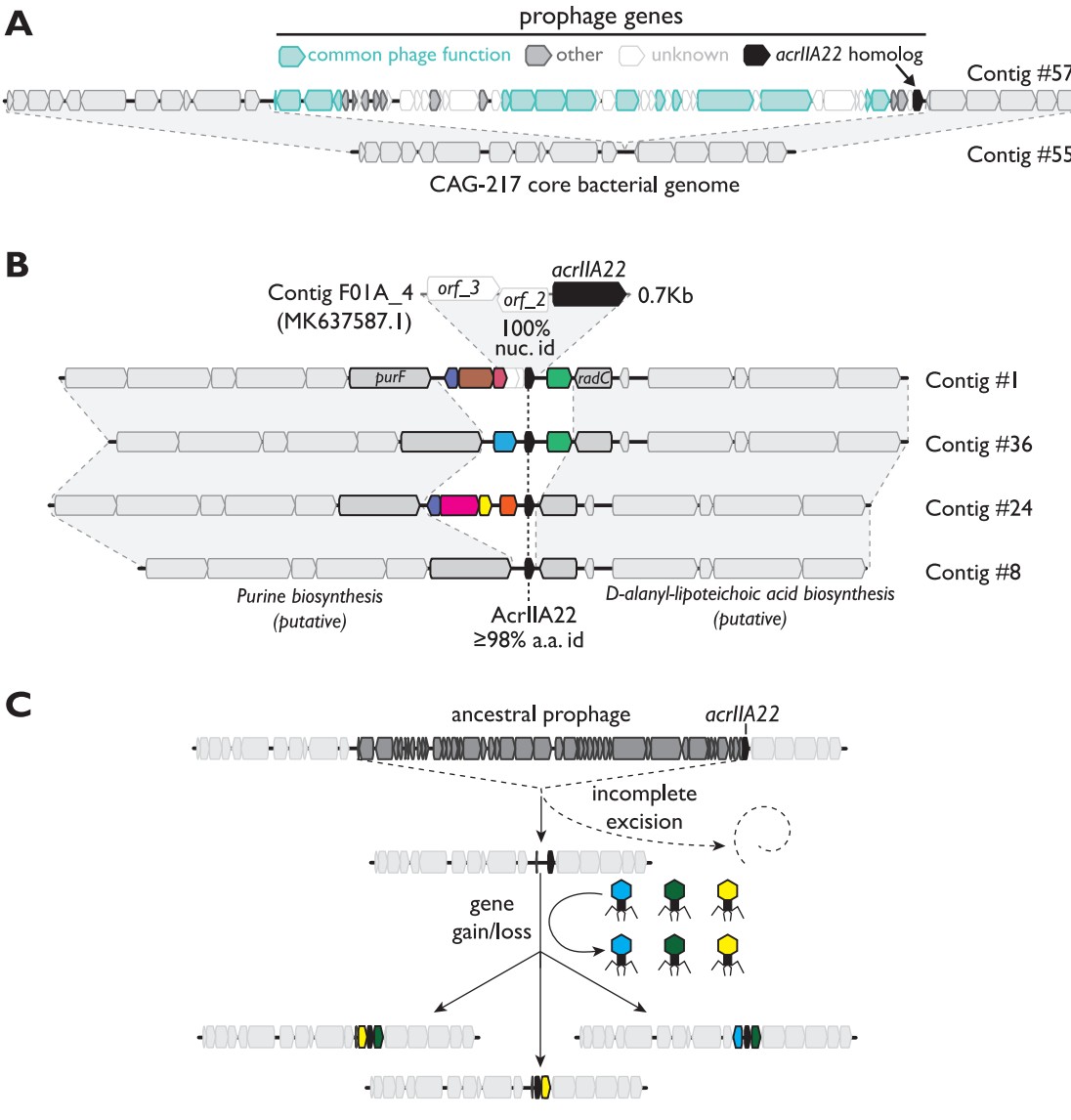

**Fig 2. *AcrIIA22* homologs are found in hypervariable regions of prophage and bacterial genomes in the unnamed clostridial genus, CAG-217.** (**A**) We show a schematic representation of an *acrIIA22* homolog embedded in a prophage genome, which is integrated into a bacterial genome (contig #57). We can delineate precise boundaries of the inserted prophage based on comparison to a near-identical bacterial contig (contig #55). Prophage genes are colored by functional category, according to the legend at the top. Bacterial genes are colored light gray. (**B**) Homologs of *acrIIA22* are depicted in diverse genomic islands, including Contig #1, whose sequence includes a portion identical to F01A_4, the original metagenomic contig we recovered. All *acrIIA22* homologs in these loci are closely related, but their adjacent genes are different, unrelated gene families (depicted by different colors). Genomic regions flanking these hypervariable islands, including genes immediately adjacent to these islands (*purF* and *radC*, in bold outlines), are nearly identical to one another (≥98% nucleotide identity). Contigs are numbered to indicate their descriptions in S3 Table, which contains their metadata, taxonomy, and sequence retrieval information. All sequences and annotations may also be found in S2 and S3 Data. (**C**) We propose an evolutionary model for the origin of the *acrIIA22*-encoding hypervariable genomic islands depicted in panel B. We propose that *acrIIA22* moved via prophage integration into a bacterial genomic locus but remained following an incomplete prophage excision event. Its neighboring genes subsequently diversified via horizontal exchange with additional phage genomes without these phage genomes inserting into the locus. S4 Fig depicts a more detailed version of the genomic data underlying this model.

In total, we identified 30 unique *acrIIA22* homologs, 25 of which were predicted to originate from the unnamed clostridial genus, CAG-217 (Fig 3A). Because *acr*s are only beneficial to phages if they inhibit CRISPR-Cas activity, they are typically found only in taxa with a high prevalence of susceptible Cas proteins [9]. If AcrIIA22 functions naturally as an Acr, we would predict that Cas9-encoding, type II-A CRISPR-Cas systems like SpyCas9 would be common in CAG-217 bacteria. To test this idea, we examined 779 draft assemblies of CAG-217 genomes and found that 179 of the 181 predicted CRISPR-Cas systems were type II-A systems (the remaining 2 loci were Cas12-encoding, type V-A systems). This enrichment for Cas9 is particularly striking as *Clostridia* typically encode other CRISPR-Cas defenses and only rarely encode Cas9-based systems [21]. Moreover, prophages from CAG-217 encode 78 type II-A Acrs (homologs of AcrIIA7, AcrIIA17, and AcrIIA21), suggesting that they are actively engaged in an arms race with Cas9-based defenses in these bacteria. In one case, we found *acrIIA17* and *acrIIA22* homologs within 1 kilobase of each other in a prophage genome (S5 Fig) [22]. Phages often collect *acr* genes in the same genomic locus [23], commonly pairing narrow-spectrum *acr*s that act during lytic infection alongside broad-spectrum *acr*s that operate during lysogeny [24]. Together, these observations support our hypothesis that prophages encode *acrIIA22* homologs to inhibit type II-A CRISPR-Cas (Cas9) systems in CAG-217 genomes.

We next tested whether the ability to inhibit type II-A CRISPR-Cas systems was shared among *acrIIA22* homologs from CAG-217 bacteria. To do so, we selected *acrIIA22* homologs that spanned the phylogenetic diversity present among CAG-217 genomes (Fig 3A) and tested their ability to protect a target plasmid from SpyCas9 elimination. These analyses revealed that diverse *acrIIA22* homologs from CAG-217 bacteria (for example, sharing only 56.9% identity) could antagonize SpyCas9 activity at least partially (Fig 3B), reminiscent of the broad inhibition that has been previously observed for some other type II-A Acrs [12]. To determine if this Acr activity extended beyond SpyCas9, we used a slightly modified plasmid protection assay (see Methods) to test whether *acrIIA22* could inhibit other type II and type V CRISPR-Cas systems, as these were the only 2 CRISPR-Cas types present in CAG-217 genomes. Though *acrIIA22* could not inhibit any of the type V (Cas12-encoding) systems we tested, it did protect a target plasmid from 2 substantially diverged type II CRISPR-Cas systems, consistent with the high prevalence of Cas9-based systems among CAG-217 bacteria (Fig 3C). Such broad-spectrum inhibition can occur either by targeting a conserved feature of Cas9 or by inhibiting Cas9 via an indirect mechanism that it cannot easily evade.

## AcrIIA22 functions via a noncanonical mechanism

Almost all characterized Acrs inhibit their cognate Cas proteins via direct binding without the involvement of additional cofactors. As a result, they exhibit strong inhibitory activity when tested in vitro (S1 Table). To determine if this was the case for AcrIIA22, we purified it from *E. coli* and asked whether it could bind and inhibit SpyCas9. To test for binding, we asked whether twin-strep-tagged AcrIIA22 coprecipitated with untagged SpyCas9 when mixed as purified proteins. Unlike with AcrIIA4, which binds strongly to SpyCas9 and inhibits its activity in vitro, we detected little to no binding between AcrIIA22 and SpyCas9, regardless of whether a single-guide RNA (sgRNA) was included or not (S6 Fig). We also observed that AcrIIA22 had no impact on SpyCas9's ability to cleave linear, double-stranded DNA (dsDNA), even when AcrIIA22 was included at substantial molar excess over SpyCas9 (S7 Fig). These results suggest that AcrIIA22 cannot bind and inhibit SpyCas9, at least in isolation. Thus, AcrIIA22 lacks the predominant biochemical activities exhibited by previous Acrs that have been mechanistically characterized.

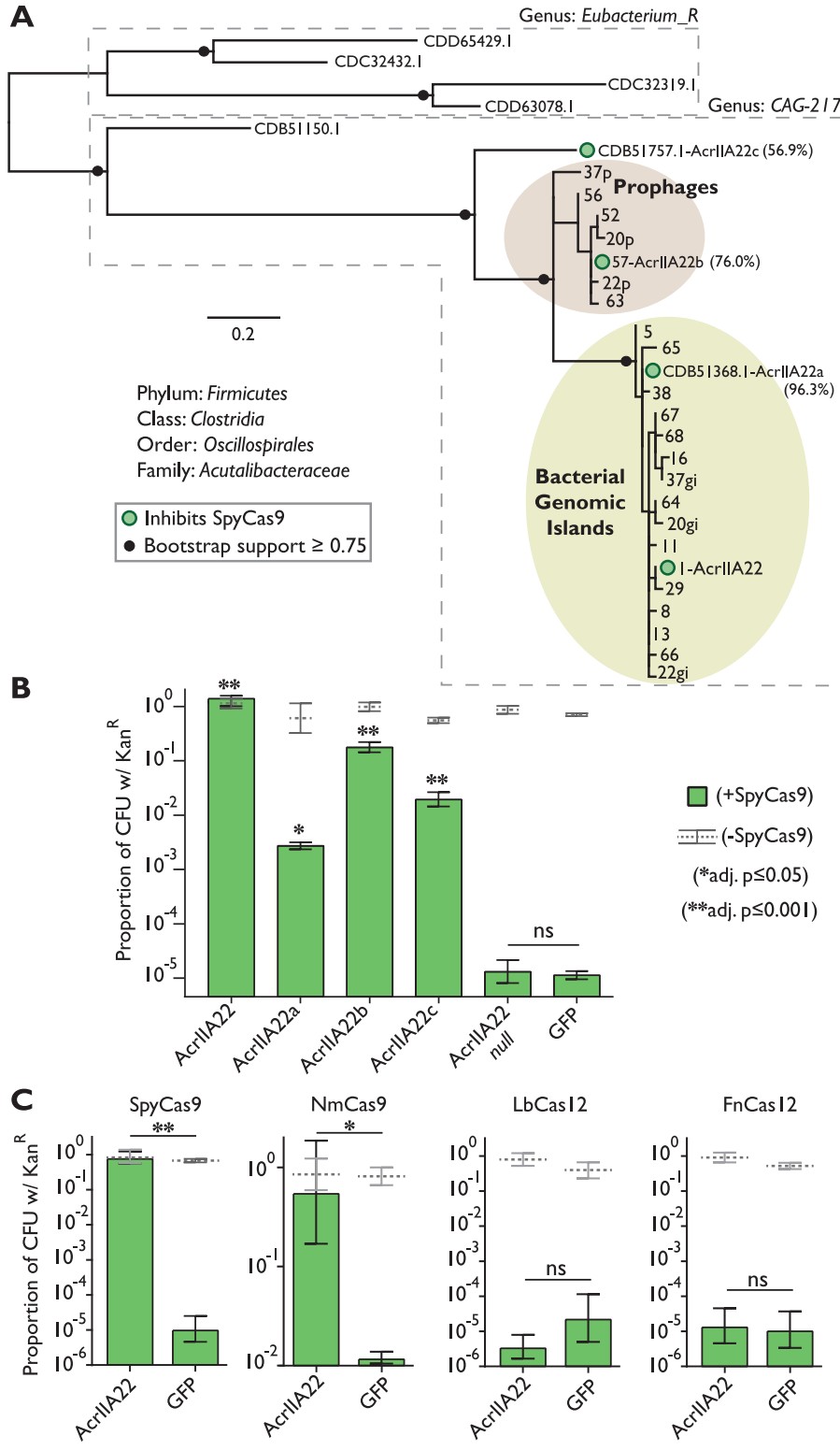

**Fig 3. Several AcrIIA22 homologs in the CAG-217 clostridial genus can inhibit SpyCas9.** (**A**) A phylogeny of all unique AcrIIA22 homologs identified from metagenomic and NCBI databases. Phylogenetic classifications were assigned corresponding to the GTDB naming convention (Methods). Prophage sequences are shaded brown and homologs from hypervariable bacterial genomic islands are shaded yellow. Sequences obtained from NCBI are labeled with protein accession numbers. In other cases, AcrIIA22 homologs are numbered to match their contig-of-origin (S3

Table). In some cases, more than one AcrIIA22 homolog is found on the same contig ("gi" or "p" indicates its presence in a hypervariable genomic island or prophage genome, respectively). Circles at nodes indicate bootstrap support ≥ 0.75. Dashed boxes separate sequences identified from CAG-217 versus *Eubacterium_R* bacterial genera. Filled green circles indicate homologs that were tested for their ability to inhibit SpyCas9 in the plasmid protection assay in panel B. These homologs have been named with "a," "b," or "c" suffixes to distinguish them from the original AcrIIA22 metagenomic hit; their amino acid identity to the original hit is shown in parentheses. (**B**) Several homologs of AcrIIA22 in CAG-217 genomes inhibit SpyCas9. Asterisks depict statistically significant differences in plasmid retention under SpyCas9-inducing conditions between the indicated sample and a null mutant with an early stop codon in *acrIIA22*, as indicated in the legend at right (ns indicates no significance; $p > 0.05$). All *p*-values were corrected for multiple hypotheses using Bonferroni method (Student *t* test, $n = 3$). (**C**) AcrIIA22 inhibits divergent Cas9 proteins from *Streptococcus pyogenes* (SpyCas9) or *Neisseria meningitidis* (NmCas9) but not Cas12 proteins from *Lachnospiraceae bacterium* (LbCas12) or *Francisella novicida* (FnCas12). As in panel B, green bars indicate samples with expression of the indicated Cas nuclease, while unexpressed controls are depicted with gray lines. For Cas-expressing samples, significance was determined via a Student *t* test ($n = 3$) and denoted as follows: "$*$," $p \leq 0.05$; "$**$" $p \leq 0.001$; "ns" no significance. Due to slight differences in the plasmid protection assay in panel C compared to panel B, A22 was retested against SpyCas9 to confirm activity (Methods). The individual numerical values that underlie the summary data in this figure may be found in S1 Data. GTDB, genome taxonomy database; SpyCas9, *Streptococcus pyogenes* Cas9.

We therefore considered the possibility that AcrIIA22 encodes an unconventional Acr that acts via a noncanonical mechanism. However, the only AcrIIA22 homologs we could detect using BLAST were proteins of unknown function, which provided few clues about AcrIIA22 activity or biochemical mechanisms. Anticipating that structural homology might provide better insight into its mechanism of inhibition, we solved AcrIIA22's structure using X-ray crystallography. We first built a de novo model from AcrIIA22's primary sequence with Robetta [25]. We then used this model as a molecular replacement probe to solve its structure at 2.80 Å resolution (PDB:7JTA). The asymmetric unit in AcrIIA22's crystal comprises 2 monomers stacked end to end, with each monomer folding into a 4-stranded β-sheet (Fig 4A and Table 1). A DALI structure–structure search revealed that the AcrIIA22 monomer is similar to members of the newly recognized PC4-like structural fold (Fig 4B and S2 Table). PC4-like proteins have independently evolved in all domains of life, typically adopt a β-β-β-β-α topology, and often homodimerize to bind diverse RNA and DNA species using variably positioned β-sheets [26].

Despite crystallizing as a homodimer, AcrIIA22 migrated from a size exclusion chromatography (SEC) column at an elution volume corresponding to a calculated mass approximately 4 times larger than its expected monomeric molecular weight (Fig 4B). This suggested that AcrIIA22 may oligomerize in vivo. Indeed, AcrIIA22 was predicted to form a stable tetramer when analyzed with PISA, a tool for inferring macromolecular assembles from crystal structures [27] (Fig 4C and S8A and S8B Fig). This putative tetramer has a molecular mass consistent with that observed by SEC and comprises pairs of outward-facing, concave β-sheets. A series of hydrophobic interactions likely stabilize this configuration of β-sheets instead of the typical α-helical interactions seen in other PC4-like proteins, potentially explaining the absence of an α-helix in AcrIIA22 (S8C and S8D Fig). Interestingly, many PC4-like proteins bind nucleic acids using similar concave β-sheets and, in some instances, form higher-order oligomers as an obligate step for binding and/or unwinding DNA or RNA [26]. Consistent with this possibility, AcrIIA22's β-sheets orient along each outward face of the putative tetramer, resemble those in PC4-like proteins, and form a groove that could potentially accommodate a nucleic acid substrate (Fig 4D and S8A, S8B and S8E Fig). Thus, AcrIIA22's structural and functional attributes led us to suspect that it could also interact with nucleic acids and potentially affect their topology.

Our tetramer model predicts that an interface at the C-terminus of AcrIIA22 is required for adjacent β-sheets to bind one another and form a grooved, oligomeric structure (Fig 4C and

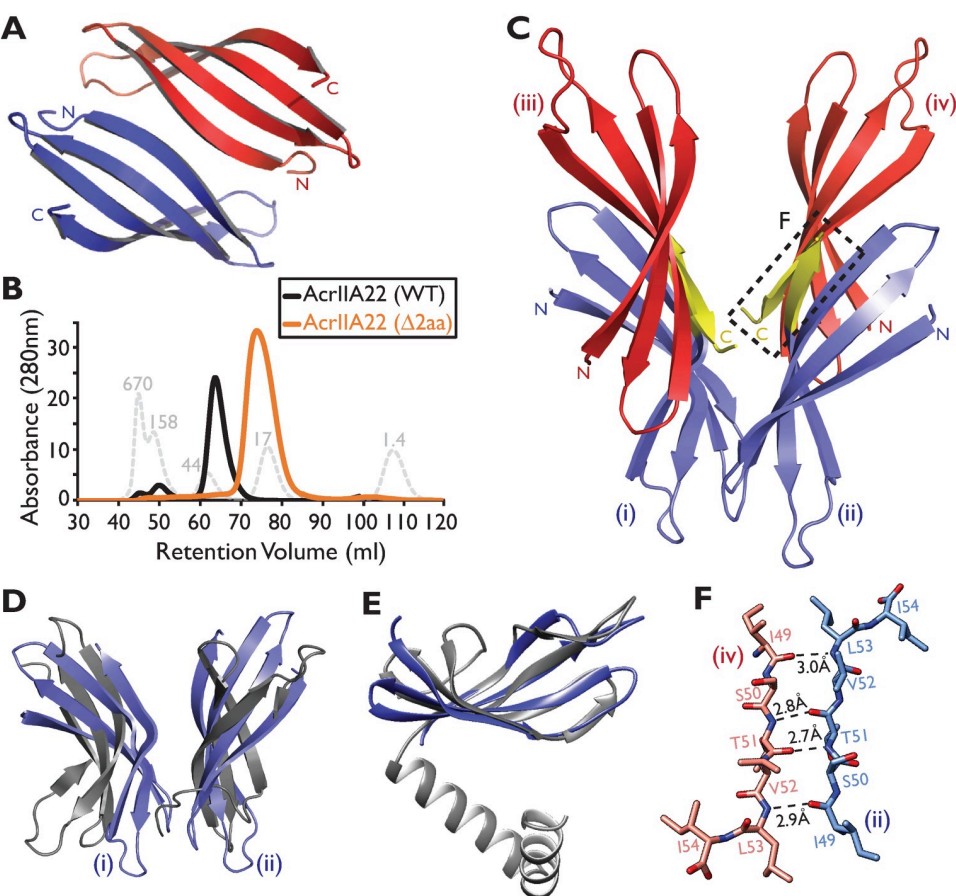

**Fig 4. AcrIIA22 is an oligomeric PC4-like protein.** (**A**) AcrIIA22's crystal structure reveals a homodimer of 2 four-stranded β-sheets. (**B**) AcrIIA22 elutes as an oligomer that is approximately 4 times the predicted molecular mass of its monomer, which is 7 kDa. The gray, dashed trace depicts protein standards of the indicated molecular weight, in kDa. The orange trace depicts the elution profile of a 2-amino acid C-terminal AcrIIA22 truncation mutant that is predicted to disrupt oligomerization. (**C**) Ribbon diagram of a proposed AcrIIA22 tetramer, which requires binding between antiparallel β-strands at the C-termini of AcrIIA22 monomers to form extended, concave β-sheets. The putative oligomerization interface is indicated by the regions highlighted in yellow and the dashed box and is detailed further in panel F. Each monomer in the proposed tetramer is labeled with lowercase Roman numerals (i-iv). (**D**) β-sheet topology and orientation in AcrIIA22 (blue) resemble that of PC4-like family proteins (in gray, PDB:4BG7 from phage T5). (**E**) A monomer of AcrIIA22 (in blue, PDB:7JTA) is structurally similar to a PC4-like single-stranded DNA binding protein, which is proposed to promote recombination in phage T5 (in gray, PDB:4BG7, Z-score = 6.2, matched residues 15%), except for a missing C-terminal alpha helix. (**F**) A putative oligomerization interface between the C-termini of 2 AcrIIA22 monomers from panel (C) is shown in more detail. Dashed lines indicate potential hydrogen bonds between the polypeptide backbones. This interface occurs twice in the putative tetramer, between red-hued and blue-hued monomers in panel C.

4F). We reasoned that a 2-residue, C-terminal truncation of AcrIIA22 would disrupt this interface (Fig 4F and S8G Fig). To test this hypothesis, we examined the oligomeric state of this 2-aa AcrIIA22 deletion mutant by SEC. Consistent with our hypothesis, we found that the mutant AcrIIA22 complexes migrated at half the size of the wild-type complexes, corresponding to approximately twice AcrIIA22's molecular weight (Fig 4B). This suggests that the C-terminal interface is required to progress from a 2- to 4-membered oligomer, consistent with our model. Moreover, we found that the 2-aa deletion mutant was also impaired for SpyCas9 antagonism in our plasmid protection assay (S9A Fig). Thus, this C-terminal motif is necessary

**Table 1. Structural features of AcrIIA22.**

| Data collection | |
|---|---|
| Space Group | P4332 |
| *Cell Dimensions* | |
| a, b, c (Å) | 128.56, 128.56, 128.56 |
| α, β, γ (˚) | 90.0, 90.0, 90.0 |
| Resolution (Å) | 50.00–2.80 |
| $R_{merge}$ | 0.106 (0.906) |
| $I/\sigma_I$ | 17.4 (2.6) |
| Completeness (%) | 98.7 (100.0) |
| Redundancy | 10.4 (10.7) |
| CC 1/2 | 0.837 |
| **Refinement** | |
| No. Reflections | 9,334 |
| $R_{work}$ ($R_{free}$) (%) | 22.2 (24.6) |
| No. Complex in ASU | 2 |
| *No. atoms* | |
| Protein | 810 |
| Heteroatoms | 50 |
| Water | 3 |
| B-factor | 82.82 |
| *R.m.s deviations* | |
| Bond lengths (Å) | 0.003 |
| Bond angles (˚) | 0.610 |
| *Ramachandran* | |
| Preferred (%) | 98.15 |
| Allowed (%) | 1.85 |
| Outliers (%) | 0 |

for protection from SpyCas9 and for higher-order oligomerization, suggesting that oligomerization may be necessary for AcrIIA22's anti-SpyCas9 activity.

## AcrIIA22 is a DNA nickase that relieves torsion of supercoiled plasmids

Our structural analyses indicated that AcrIIA22 is a PC4-like nucleic acid–interacting protein. Like AcrIIA22, many of the known PC4-like proteins are encoded in phage genomes. Among these is AcrIIA22's closest structural relative in the PC4-like family: a predicted single-stranded binding (SSB) protein from phage T5 (Fig 4E) [28]. This putative SSB protein has been predicted to directly stimulate recombination during the recombination-dependent replication of phage T5's genome [29]. This prediction, together with our inference from genomic analyses (Fig 2 and S4 Fig), led us to hypothesize that AcrIIA22 may have similar recombination-stimulating activity. Indeed, other PC4-like proteins have been observed experimentally to unwind duplex DNA, a function consistent with their proposed roles in transcription and recombination [26,30]. Therefore, we investigated whether AcrIIA22 might also similarly interact with duplexed DNA to affect its topology.

We investigated whether we could detect any biochemical effect of *acrIIA22* on a dsDNA plasmid in vivo. In this experiment, we compared 2 *acrIIA22* genotypes: the wild-type sequence and a null mutant with a single base pair change to create an early stop codon. We grew overnight cultures of plasmids expressing each genotype, purified plasmid DNA, and

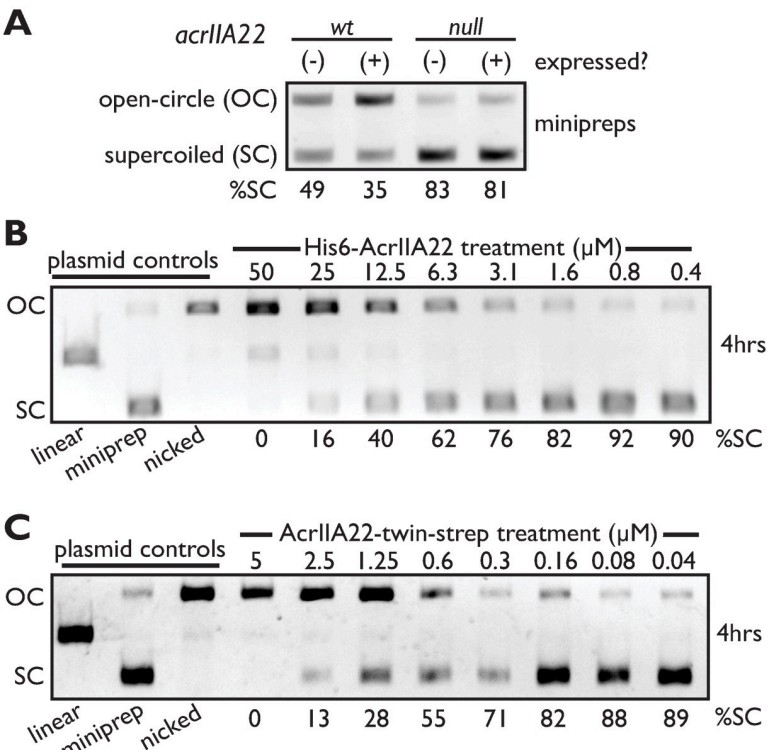

**Fig 5. AcrIIA22 nicks supercoiled plasmids in vivo and in vitro.** (**A**) Gel electrophoresis of plasmids purified from overnight *E. coli* cultures expressing either acrIIA22, or a null mutant with an early stop codon, or neither. Compared to the null mutant, more plasmid runs in a slowly migrating, OC conformation rather than SC plasmid with the wild-type *acrIIA22* allele, suggesting that *acrIIA22* may impact plasmid topology. %SC indicates the percentage of DNA in the SC form for each sample. (**B**) N-terminally His6-tagged AcrIIA22 nicks supercoiled plasmids in vitro. (**C**) C-terminally twin-strep-tagged AcrIIA22 nicks supercoiled plasmids in vitro with higher specific activity than shown in panel B (compare protein concentrations). Original, uncropped versions of images depicted in figure may be found in the Supporting information file, S1 Raw Images. OC, open-circle; SC, supercoiled.

analyzed its topology using gel electrophoresis (Fig 5A). As is typical for plasmid purifications from *E. coli*, the plasmid encoding the null mutant was predominantly recovered in a super-coiled form. In contrast, AcrIIA22 expression shifted much of the target plasmid to a slowly migrating form, consistent with an open-circle conformation. These findings suggest that AcrIIA22 expression could relieve plasmid supercoiling, potentially via DNA nicking activity.

Though *acrIIA22* expression appeared to alter plasmid topology in vivo, DNA topology is a dynamic process, regulated by many competing factors and dependent on cellular physiology [31]. Thus, we could not attribute the observed change in plasmid conformation solely to AcrIIA22. To more directly investigate AcrIIA22's effect on plasmid topology, we purified an N-terminal, His6-tagged AcrIIA22 protein and examined its impact on a plasmid DNA sub-strate in vitro. By gel electrophoresis, we observed that AcrIIA22 shifted a supercoiled plasmid to a slowly migrating form in a time and concentration-dependent manner (Fig 5B and S10D Fig). For comparison, we also treated a plasmid with the nickase Nb.BssSI, yielding a band that migrated at the same position as the putatively open-circle product generated via AcrIIA22 activity (Fig 5B). High concentrations of AcrIIA22 resulted in conversion of plasmids to a line-arized DNA product, consistent with a nickase-like nuclease activity acting on both strands of DNA (Fig 5B). This nicking activity was strongly stimulated in the presence of $Mn^{2+}$, $Co^{2+}$, and $Mg^{2+}$, weakly with $Ni^{2+}$ and $Zn^{2+}$, and not at all with $Ca^{2+}$ (S11 Fig). To confirm that the

observed gel shift was the result of nicking activity and not protein-bound DNA, we purified an AcrIIA22-treated plasmid with phenol-chloroform and reexamined it by gel electrophoresis. We observed that the open-circle form of the plasmid persisted through purification, establishing it as the product of a bona fide nickase (S11 Fig). Consistent with our in vivo observations (S9A Fig), we found that the 2-aa deletion mutant was impaired for nicking activity relative to wild-type AcrIIA22 (S9B Fig). These data suggested that *acrIIA22* may encode for a protein that nicks DNA.

No known nuclease has been previously characterized among the PC4-family proteins [26]. Therefore, to further test our hypothesis that AcrIIA22 nicks supercoiled plasmids, we performed several additional experiments. First, we repurified an N-terminal, His6-tagged AcrIIA22 protein, but this time examined individual fractions for nicking activity. Consistent with AcrIIA22's hypothesized function, nicking activity correlated with AcrIIA22 concentration across these fractions (S10B and S10C Fig); no copurifying contaminant was detected via Coomassie stain (S10A Fig). This nicking activity, however, was low enough that we could not eliminate the possibility that another protein, undetectable via Coomassie stain, might have copurified with AcrIIA22 and could explained this behavior. Reasoning that different contaminating proteins would result from different purification strategies, we generated a new version of the AcrIIA22 protein and purified it via a C-terminal, twin-strep-tag. A more sensitive, silver-stained gel indicated that this AcrIIA22 preparation was also very pure (S10E Fig). We subsequently confirmed that it nicked supercoiled plasmids with a specific activity of $5.1 \times 10^{-7}$ nmol/min/mg (Fig 5C and S10G and S10H Fig). This activity is comparable to other nickases involved in phage-bacterial conflicts (including SspB, which nicks at a rate of $8.9 \times 10^{-7}$ nmol/min/mg) [32]. Notably, this specific activity is significantly higher than we observed for our original, N-terminal His6-tagged variant (compare AcrIIA22 concentrations in Fig 5B and 5C). This difference in nicking activity is also reflected in plasmid protection phenotypes observed in vivo; only C-terminally tagged AcrIIA22, but not N-terminally tagged AcrIIA22, protected a plasmid from SpyCas9 attack (S10F Fig). Thus, our studies find a strong correlation between AcrIIA22's nicking and plasmid protection activities.

If AcrIIA22 encoded a true nickase, we hypothesized that we might be able to abrogate this activity via point mutations in putative catalytic residues. Therefore, we searched for individual point mutations that impaired nicking activity in vitro. If such mutants existed, they would allow us to test our hypothesis that AcrIIA22 is a nickase. Reasoning that acidic amino acids were likely to be important catalytic residues [33], we individually changed each aspartic acid and glutamic acid in AcrIIA22 to an alanine. Hypothesizing that AcrIIA22's in vitro biochemical activity would correlate with its anti-Cas9 function in vivo, we tested whether these alanine variants still inhibited SpyCas9 in our plasmid protection assay. Of the 11 mutants tested, D14A stood out. This mutant showed clear SpyCas9-dependent plasmid loss, with a >250-fold reduction in plasmid retention compared to a wild-type AcrIIA22 control (Fig 6A).

Purification of the D14A mutant (via a C-terminal twin-strep tag) revealed that it displayed similar expression level, purification yield, oligomeric size distribution, and solution behavior as wild-type AcrIIA22 (Fig 6B and S10E Fig), indicating that the mutant protein is still properly folded. The D14A mutant was substantially impaired for nicking activity compared to the wild-type AcrIIA22 protein (Fig 6C and S10G Fig), consistent with its diminished anti-Cas9 activity in vivo (Fig 6A). Unlike previous observations with the 2-aa deletion mutant (Fig 4B), the reduction in nicking for the D14A mutant is unlikely to be the result of oligomeric differences between it and wild-type AcrIIA22 (Fig 6B). Instead, we speculate that D14 may contribute to AcrIIA22's nicking activity, as 2 D14 residues from different AcrIIA22 monomers sit very near to one another in our proposed tetramer, such that they may be stabilized via the presence of a divalent cation under physiological conditions (S8F Fig).

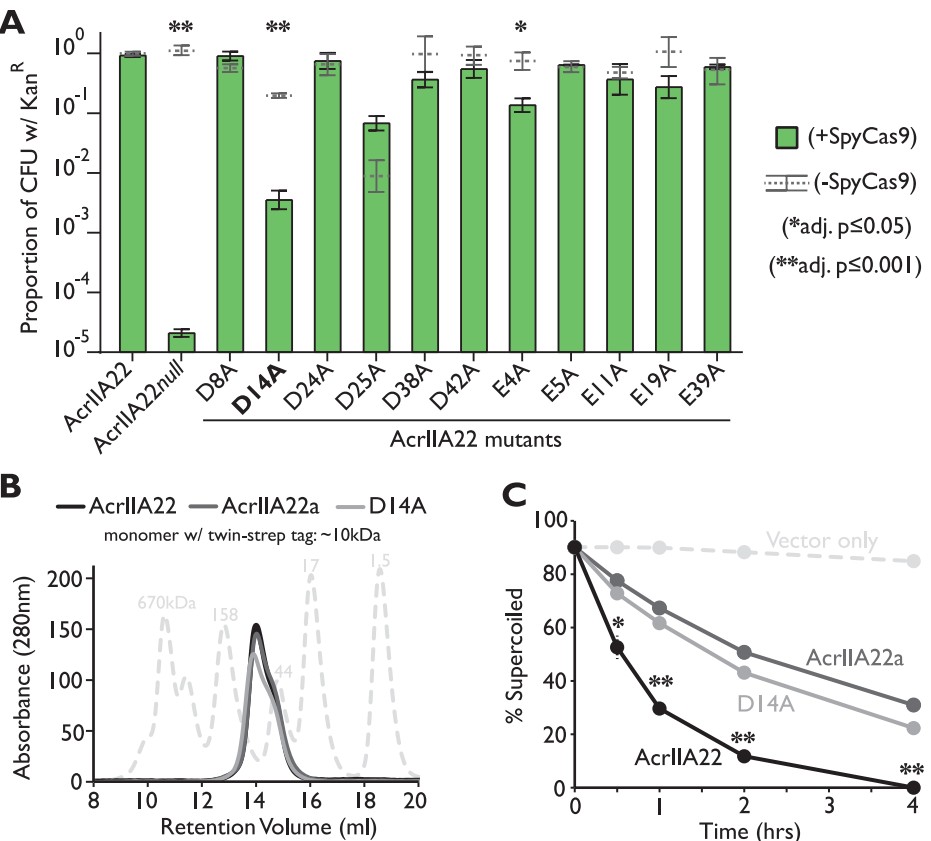

**Fig 6. Impaired nicking activity of AcrIIA22 variants in vitro correlates with lower SpyCas9 inhibition in vivo.** (**A**) Alanine mutagenesis of acidic amino acid residues (glutamic acid or aspartic acid) in AcrIIA22 reveals that D14 is important for plasmid protection against SpyCas9. Asterisks depict statistically significant differences in plasmid retention under SpyCas9-inducing and noninducing conditions, per the legend at right. The D14A mutant is significantly impaired, the E4A mutant is slightly impaired, whereas all other mutants are not impaired for plasmid protection against SpyCas9 compared to an uninduced control. All *p*-values were corrected for multiple hypotheses using Bonferroni method (Student *t* test, *n* = 3). (**B**) AcrIIA22 (black), AcrIIA22a (dark gray), and a D14A mutant (light gray) all elute with similar oligomer profiles via SEC. The dashed trace depicts protein standards of the indicated molecular weight, in kDa. (**C**) AcrIIA22a and the D14A mutant are impaired for nicking relative to AcrIIA22. All experiments were performed in triplicate, with standard deviations indicated by dashed lines (in most cases, the data points obscure these error bars). Asterisks denote cases where AcrIIA22 is significantly different than both AcrIIA22a and the D14A mutant after correcting for multiple hypotheses (Student *t* test, *n* = 3, Bonferroni correction). A single asterisk (\*) means that adjusted *p*-values for both comparisons are below 0.05. A double asterisk (\*\*) means that adjusted *p*-values are both below 0.005. S10G and S10H Fig show representative gels for these nicking experiments. The individual numerical values that underlie the summary data in this figure may be found in S1 Data. CFU, colony-forming unit; SEC, size exclusion chromatography; SpyCas9, *Streptococcus pyogenes* Cas9.

Our surveys of divergent AcrIIA22 homologs also revealed a naturally occurring AcrIIA22 homolog with diminished function in vivo (AcrIIA22a, Fig 3B). Despite encoding for a protein that differs by only 2 amino acids from the original sequence (V3I and R30K), *acrIIA22a* was >450-fold less effective at protecting a plasmid from SpyCas9 than *acrIIA22* (Fig 3B). We examined whether this loss of SpyCas9 protection correlated with loss of nicking activity, just like the D14A mutant. Upon purification, a twin-strep-tagged AcrII22a protein eluted with an SEC profile identical to that of AcrIIA22, suggesting a similar oligomeric state (Fig 6B). Yet, AcrIIA22a exhibited substantially less nicking activity that wild-type AcrIIA22 in vitro (Fig 6C). In our proposed AcrIIA22 tetramer, R30 likely forms a hydrogen bond with the C-terminus of a diagonal monomer, raising the possibility that the R30K variant alters the protein's

conformational plasticity or mediates other allosteric effects (S8G Fig). As with D14A, the partial loss of nicking activity seen for AcrIIA22a (Fig 6C) correlated with a partial loss of plasmid protection against SpyCas9 (Fig 3B). Thus, we describe 2 closely related AcrIIA22 variants, one engineered and one naturally occurring, whose nicking activity in vitro corresponds directly to plasmid protection in vivo. From these data, along with our other in vitro and in vivo findings, we conclude that *acrIIA22* encodes a nickase protein that relieves the torsional stress of supercoiled plasmids.

## AcrIIA22's nicking activity indirectly impairs SpyCas9

Having established that AcrIIA22 is a DNA nickase, we next investigated how this biochemical activity may enable its inhibition of SpyCas9 without directly binding the Cas protein. We therefore tested the consequences of expressing AcrIIA22 on a target plasmid in the presence of SpyCas9. As before, we began by comparing overnight plasmid purifications of a target plasmid expressing AcrIIA22 and a null mutant with an early stop codon as a negative control. For both genotypes, we subjected the *acrIIA22*-encoding plasmid to SpyCas9 targeting during bacterial growth. We were unable to recover the negative control target plasmid after overnight growth, implying that this target plasmid was eliminated by SpyCas9 (Fig 7A). In contrast, SpyCas9 did not eliminate a target plasmid that expressed full-length AcrIIA22 (Fig 7A), consistent with AcrIIA22's capacity to protect against SpyCas9 (Fig 1C).

To be effective, a CRISPR-Cas system must eliminate its target at a faster rate than the target can replicate [34]. Our findings raised the possibility that AcrIIA22 modifies a target plasmid into a SpyCas9-resistant conformation to win this "kinetic race" against SpyCas9, potentially shifting the equilibrium to favor plasmid persistence instead of elimination. To test this kinetic race model, we asked whether a plasmid that had been pretreated with AcrIIA22 could resist digestion by SpyCas9 in vitro. Therefore, we purified the open-circle plasmid that resulted from AcrIIA22 pretreatment and determined how efficiently it was cleaved by SpyCas9 compared to an unmodified, supercoiled plasmid (Fig 7B). SpyCas9 showed a clear preference for cleaving the supercoiled substrate versus the AcrIIA22-treated open-circle plasmid (Fig 7C–7E), consistent with previous reports [13–16]. An open-circle plasmid pretreated with the nickase Nb.BssSI was similarly recalcitrant to SpyCas9 digestion (Fig 7C and 7D). Taken together, our findings suggest that relieving DNA torsion provides the mechanistic explanation for AcrIIA22's ability to inhibit SpyCas9 in vivo.

## Discussion

In this study, we identify and characterize *acrIIA22*, a previously undescribed gene that can antagonize SpyCas9. We show that AcrIIA22 homologs are common in genomes and prophages of CAG-217 bacteria, which have a high prevalence of Cas9 homologs. Using a combination of structural and biochemical studies, we show that AcrIIA22 acts by nicking supercoiled DNA to relieve torsional stress on a target plasmid and that this activity correlates with protection against SpyCas9 in vivo and in vitro. This torsion-based model for SpyCas9 inhibition helps explain why AcrIIA22 protects plasmids more effectively than phage Mu against SpyCas9. Because plasmids are maintained as circular, extrachromosomal elements, they are more likely to undergo torsional change when nicked than the dsDNA genome of phage Mu, which is injected as linear DNA and spends significant time integrated into *E. coli*'s genome [35]. Additionally, linear DNA experiences much lower torsional stress and therefore is less susceptible than supercoiled plasmids to cleavage by SpyCas9 [15]. This difference also likely explains why AcrIIA22 failed to protect a linear dsDNA substrate from SpyCas9 during our earlier in vitro experiments (S7 Fig).

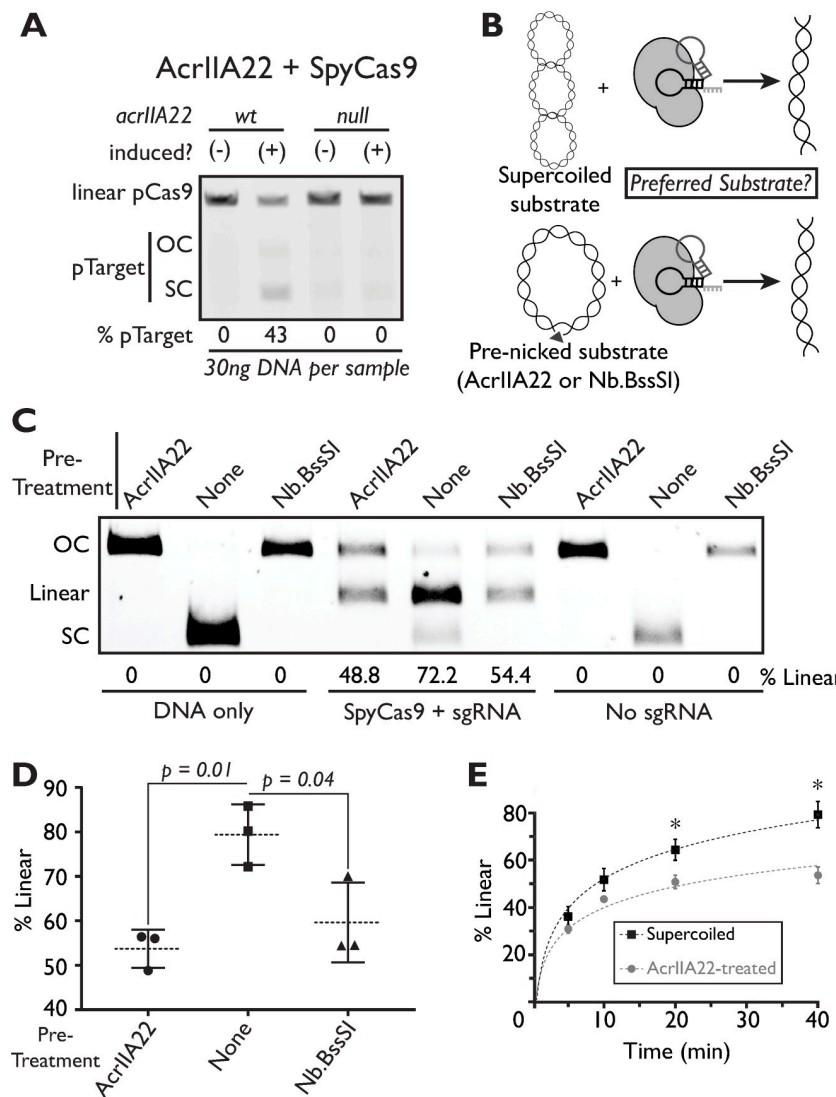

**Fig 7. Nicking by AcrIIA22 protects plasmids from SpyCas9 in vivo and in vitro.** (**A**) Gel electrophoresis of plasmids purified from overnight *E. coli* cultures expressing either wild-type *acrIIA22* or a mutant with an early stop codon ("null"). In these cultures, SpyCas9 was expressed from a second plasmid, which was linearized via a unique restriction site before electrophoresis. The *acrIIA22*-encoding plasmids are indicated with the "pTarget" label. The "% pTarget" figure indicates the fraction of total DNA attributable to pTarget, quantified by densitometry analysis. In cases with complete pTarget elimination, all DNA comes from the SpyCas9 expression plasmid, and thus these bands are more pronounced. However, in the presence of wild-type acrIIA22, pTarget is protected from SpyCas9-mediated cleavage and makes up 43% of total plasmid DNA. (**B**) We present a schematic of the experimental design for the data depicted in panel C. The experiment tests whether SpyCas9 preferentially cleaves an SC or OC plasmid target in vitro. Though both plasmid substrates will be linearized following SpyCas9 cleavage, linear DNA will accumulate more readily with a preferred substrate. (**C**) Plasmid purifications from overnight cultures were either left unmodified or pretreated with one of 2 nickase enzymes, AcrIIA22 or Nb.BssSI, following which each substrate was digested with SpyCas9 in vitro. The percentage of DNA in the linear form is quantified below the gel, which indicates complete SpyCas9 cleavage. Linear, OC, and SC plasmid forms are indicated along with the left of the gel and reaction components below the gel. SpyCas9 cuts DNA strands sequentially; incomplete digestions with supercoiled substrates produce OC plasmids if only one strand has been cleaved (e.g., lane 5). Prenicked plasmids, by either AcrIIA22 or Nb. BssSI, are less susceptible to linearization via SpyCas9 cleavage. (**D**) Endpoint measurements indicate that SpyCas9 more efficiently linearizes SC plasmids than substrates nicked with either AcrIIA22 or Nb.BssSI (Student *t* test, *n* = 3). (**E**) A time course experiment demonstrates that SpyCas9 more efficiently linearizes supercoiled plasmids than AcrIIA22-treated substrates. An asterisk (*) denotes significant differences between AcrIIA22-treated and untreated substrates (Student *t* test, *p* < 0.05, *n* = 3). The individual numerical values and original images for the data presented in this figure may be found in S1 Data and S1 Raw Images, respectively. OC, open-circle; SC, supercoiled; SpyCas9, *Streptococcus pyogenes* Cas9.

Previous in vitro experiments indicate that Cas9 requires a higher degree of negative supercoiling than type I CRISPR-Cas systems to provide the free energy needed for R-loop formation [13]. Similarly, in vivo observations have shown that DNA supercoiling promotes the recruitment of SpyCas9 to its target site in bacteria [14]. Based on these published findings, we speculate that Cas9 may be particularly susceptible to changes in DNA torsion among CRISPR-Cas systems. Thus, factors that modify DNA torsion, like AcrIIA22, could provide a general means to protect against Cas9 or other enzymes with a strong preference for negative supercoils.

Taken together, our data implicate DNA topology as a new battleground in the evolutionary arms race between CRISPR-Cas systems and MGEs. Because DNA topology is dynamically regulated in phages, plasmids and other MGEs, many topology-modifying factors already exist in these genomes. Our findings suggest that at least some of these factors could have secondary effects on CRISPR-Cas activity and thus prove useful in the context of a molecular arms race [31,36]. For instance, though not studied in the context of bacterial defense systems, the fitness of phage T4 is improved via the expression of an accessory protein that modifies DNA supercoiling and the propensity of R-loops to form [37]. Other phages, such as the intrinsically Cas9-resistant phage T5 [38], incorporate regular nicks into their genome, the function of which has eluded description for over 40 years [39]. Additionally, conjugative plasmids were recently shown to evade CRISPR-Cas in *Vibrio cholerae* by the action of homologs of the recombination proteins Redβ and λExo [40]. Based on our findings, we hypothesize that phages and MGEs targeted by Cas9 exploit factors that modify DNA topology as a general tactic to evade host immunity.

Functional selections like ours are biased toward identifying genes that work well in heterologous contexts. For example, even though AcrIIA22 is encoded on the genome of a genetically intractable bacterium, we could identify it using a functional metagenomic selection for SpyCas9 antagonism in *E. coli*. Although we have characterized its activities in *E. coli* and in vitro, we cannot be certain that AcrIIA22 functions similarly in its native context. Little is known about the life cycle of native CAG-217 phages, though many dsDNA phage genomes undergo circular, topologically constrained stages during their replicative cycles [41], during which AcrIIA22 might act to specifically overcome Cas9 immunity. Alternatively, AcrIIA22 may enable Cas9 evasion as a secondary function related to some other activity. Comparative genomics (Fig 2) and structural homology to a proposed recombination-stimulating protein of phage T5 suggest a potential role for AcrIIA22 in recombination, a process that has recently been shown to promote CRISPR-Cas evasion [40].

Nevertheless, the heterologous behavior of AcrIIA22 in *E. coli* is clearly sufficient for SpyCas9 antagonism in vivo and its nicking activity can protect plasmids from SpyCas9 in vitro. Furthermore, AcrIIA22 mutants that are defective for nicking in vitro (Fig 6C and S9B Fig) are orders of magnitude less effective at protecting a plasmid from SpyCas9 in vivo (Figs 3B and 6A and S9A Fig). This indicates that modest changes in nicking activity can have major consequences for plasmid survival, which is consistent with our kinetic race model (Fig 7B) and previous observations that nonlinear equilibrium dynamics determine whether an MGE withstands CRISPR-Cas immunity [34].

Our results suggest that other proteins that affect DNA torsion may also enable Cas9 antagonism. For example, in addition to AcrIIA22, the Nb.BssSI nickase was capable of protecting a plasmid from SpyCas9 in vitro. Yet, despite the regular occurrence of nickases in nature, functional selections for anti-Cas9 activity have not previously recovered such enzymes [12,42]. We speculate that AcrIIA22 was identified from a metagenomic library because it treads a balance between activity and toxicity in *E. coli*; its nicking activity is high enough to antagonize SpyCas9 in a kinetic race but not so high that it would be toxic to the host cell (S1 Fig). Such a

balance could result from the inherent activity of the enzyme or via some form of regulation, either direct or indirect. AcrIIA22's activity is probably also regulated in its native context to avoid secondary impacts on other essential processes. Potential forms of regulation include sequence preference, oligomerization, or transient interactions with Cas9 or other host factors (Fig 4B and 4C). Studies of other phage- and bacterial-encoded nickases may provide further insight into whether AcrIIA22 proteins have additional properties that render them especially well-suited to antagonize Cas9.

Is AcrIIA22 a true Acr? AcrIIA22 lacks features that are typical of conventional Acrs, such as the ability to bind Cas proteins or to inhibit CRISPR-Cas activity as a purified protein. However, other Acr proteins also lack these features. For example, the well-characterized SpyCas9 antagonist AcrIIA1 does not inhibit purified SpyCas9 but instead stimulates Cas9 degradation in vivo [24]. Similarly, AcrIIA7 does not appear to bind SpyCas9 but can nevertheless inhibit it via an unknown mechanism in vitro [42]. Indeed, Acr proteins are defined by a common strategy and outcome rather than by a common biochemical mechanism. Our finding that *AcrIIA22* is encoded by prophages as a single gene that strongly protects plasmids and partially protects phages from SpyCas9 (Fig 3B and S3 Fig) makes it much more similar to other Acrs [23] and distinct from noncanonical CRISPR-Cas evasion strategies like DNA glucosylation [6].

Although it can protect phage Mu from SpyCas9, AcrIIA22 does not appear to provide the same potency of Cas9 inhibition as some other characterized Acrs. However, potent inhibition is not a prerequisite for effective Acr activity. In nature, multiple phages can cooperate to overcome Cas9 immunity by each contributing some Acr protein to overcome a common foe [43,44]. These dynamics can favor weak Acrs over strong ones, as the latter permits a higher incidence of cheater phages (those without Acrs) to persist in mixed phage populations [45]. Thus, even in cases where AcrIIA22 only weakly inhibits Cas9 (S3 Fig), it may nonetheless confer substantial benefit. Additionally, slowing down Cas9 cleavage could increase the time and probability for escape mutants to arise (e.g., Cas9 target-site variants [1], deletion mutants [40]), allow for additional Acr expression [43,44], or permit further genome replication to overwhelm CRISPR-Cas immunity [34]. This phenomenon—weak inhibition giving rise to long-term resistance—is reproducibly observed in cases of strong selective pressure. For instance, in the context of antibiotic resistance, the expression of QNR pentapeptide proteins by many human pathogens can provide low-level drug tolerance, extend survival, and allow time for additional mutations to develop that completely resist quinolone antibiotics [46].

As the use of functional metagenomics to study phage-bacterial conflicts grows more common, many novel genes and mechanisms for CRISPR-Cas inhibition are likely to be described [12,42]. Like AcrIIA22, which has no homology to any previously described Acr and lacks other genetic signatures used for *acr* discovery (e.g., linkage with helix-turn-helix transcription factors) [47,48], these new genes may not exhibit canonical Acr behaviors. It is inevitable that these discoveries will lead to a more nuanced understanding of the arms race between CRISPR-Cas systems and MGEs. These findings will also reveal undiscovered strategies for molecular antagonism and new battlegrounds in the age-old conflict between bacteria and their phages.

## Methods

### Plasmid protection assay

All plasmid protection assays were done in *E. coli* (strain: NEB Turbo). As described previously [12], SpyCas9 was expressed via the arabinose-inducible promoter pBAD on a CloDF13-based plasmid marked with a spectinomycin resistance cassette. The SpyCas9 construct, called

pSpyCas9_crA, was designed to eliminate a target vector with a Kan$^R$ cassette. This target vector also expressed a gene of interest (e.g., an *acr*) via the doxycycline-inducible pLtetO-1 promoter (S4 Table). We induced expression from the target vector via depression of the TetR transcription factor with doxycycline (we generically named this vector pZE21_tetR; S4 Table). IPTG was used in samples with the target vector to ensure high levels of TetR expression (which was driven by the lac promoter) and thus inducible control of our gene of interest. Unless noted in S5 Table, all genes, including each alanine mutant depicted in Fig 6A, were synthesized by Synbio Technologies and cloned directly into pZE21_tetR for functional testing.

Cultures of each sample were grown overnight at 37 ˚C with shaking at 220 rpm in lysogeny broth (LB; 10 g/L casein peptone, 10 g/L NaCl, 5 g/L ultrafiltered yeast powder) containing spectinomycin 50 μg/ml, kanamycin 50 μg/ml, and 0.5 mM IPTG. These growth conditions kept both SpyCas9 and the gene of interest in uninduced states. The next morning, overnight cultures were diluted 1:50 into LB broth containing spectinomycin (at 50 μg/ml), kanamycin (at 50 μg/ml), 0.5 mM IPTG, and doxycycline 100 ng/ml to induce the gene of interest. Cultures were grown at 37 ˚C on a roller drum to mid-log phase (for approximately 1.5 hours to OD600 of 0.3 to 0.6). Once cells reached mid-log phase, they were diluted to OD600 value of 0.01 into 2 media types: (a) LB containing spectinomycin 50 μg/ml, 0.5 mM IPTG, and doxycycline 100 ng/ml; and (b) LB containing spectinomycin 50 μg/ml, 0.5 mM IPTG, doxycycline 100 ng/ml, and 0.2% (L) arabinose. These media induced either the gene of interest alone or both the gene of interest and SpyCas9, respectively. Each sample was grown in triplicate in a 96-well plate in a BioTek Cytation 3 plate reader. After 6 hours of growth at 37 ˚C with shaking at 220 rpm, each sample was diluted 10-fold and plated on 2 types of media: (a) LB spectinomycin 50 μg/ml + 0.5 mM IPTG or (b) LB spectinomycin 50 μg/ml, kanamycin 50 μg/ml, 0.5 mM IPTG. Plates were incubated at 37 ˚C overnight. Then, colonies were counted to determine the fraction of CFUs that maintained Kan$^R$ (and thus the target vector). All figures depicting these data show the log-transformed proportion of Kan$^R$/total CFU, with or without SpyCas9 induction. The growth curves in S1 Fig match the experiment depicted in Fig 1C for the uninduced SpyCas9 samples. For the uninduced *orf_1* control samples, doxycycline was omitted from media throughout the experiment. Growth rates referenced in the text and in S1 Fig were calculated using the slope of the OD600 growth curves during log phase, following a natural log transformation.

To test AcrIIA22 function against a panel of Cas9 and Cas12 orthologs in Fig 3C, we used a slightly modified, 3-plasmid setup. As before, *spyCas9*, *nmCas9*, *fnCas12*, and *lbCas12* were encoded in a CloDF13-based plasmid with a spectinomycin resistance cassette. Expression of the Cas effector was controlled by promoter J23100 and a theophylline riboswitch. The accompanying gRNAs were encoded in a separate set of plasmids called pDual4 under an arabinose expression system, in a p15A-based plasmid and a chloramphenicol resistance cassette (S4 Table). The gRNAs in the different pDual4 constructs were programmed to target the kanamycin-marked target plasmid in the same manner as pSpyCas9_crA. All assays were done in *E. coli* (strain: NEB Turbo) following the same plasmid protection assay described previously. However, in this case, we induced expression of the different Cas effectors and gRNAs, by adding 2 mM theophylline and 0.2% (L) arabinose, respectively, to the media.

## Impact of AcrIIA22 on GFP expression

We swapped *spyCas9* for *egfp* in our CloDF13-based plasmid and coexpressed AcrIIA22 to determine if AcrIIA22 impacted expression from this construct. If AcrIIA22 influenced CloDF13's copy number or the transcription of *spyCas9*, we anticipated that it would also

impact GFP levels in this construct (pCloDF13_GFP; S4 Table). To perform this experiment, we cotransformed pCloDF13_GFP and pZE21_tetR encoding *acrIIA22* into *E. coli* Turbo. Single colonies were picked into 4 ml of LB containing spectinomycin at 50 μg/ml ("spec50") and kanamycin at 50 μg/ml ("kan50") and 0.5 mM IPTG and grown overnight at 37 ˚C shaking at 220 rpm. The next morning, the overnight culture was diluted 1:50 into both LB spec50 Kan50 + 0.5 mM IPTG with or without doxycycline (to induce *acrIIA22*) and grown at 37 ˚C for about 1.5 hours to mid-log phase (OD600 0.2 to 0.6). The OD600 was measured, and all samples were diluted to OD600 of 0.01 in 2 media types: (a) LB spec50 + kan50 + 0.5 mM IPTG + 0.2% arabinose (inducing *gfp* only) or (b) LB spec50 + kan50 + 0.5 mM IPTG + 0.2% arabinose + 100 ng/ml doxycycline (inducing *gfp* and *acrIIA22*). A volume of 200 μl of each sample was then transferred to a 96-well plate in triplicate, and GFP fluorescence was measured every 15 minutes for 24 hours (GFP was excited using 485 nm light and emission detected via absorbance at 528 nm). In parallel, we included control samples that lacked the kanamycin-marked plasmid and varied whether doxycycline was added or not (at 100 ng/ml). In these control samples, we noticed that doxycycline slightly diminished GFP expression (it is possible that sublethal levels of the antibiotic may still depress translation). Thus, we normalized GFP fluorescence measurements in our experiment with AcrIIA22 to account for this effect in all samples containing doxycycline. These normalized fluorescence measurements are shown in S2B Fig.

## Western blots to determine AcrIIA22's impact on SpyCas9 expression

We grew overnight cultures of *E. coli* Turbo that expressed pSpyCa9_crNT and pZE21_tetR encoding a gene of interest (S4 and S5 Tables) in LB spec50 + kan50 + 0.5 mM IPTG. The next morning, we diluted these cultures 1:100 in 4 ml of either (a) LB spec50 + kan50 + 0.5 mM IPTG or (b) LB spec50 + kan50 + 0.5 mM IPTG + 100 ng/ml doxycycline (to induce the gene of interest). We included samples that expressed either *acrIIA22* or *gfp* as a gene of interest. In all SpyCas9 constructs, we used a crRNA that did not target our plasmid backbone (pSpyCa9_crNT) to ensure that *acrIIA22* expression remained high and its potential impact on SpyCas9 expression levels would be most evident. All samples were grown for 2 hours at 37 ˚C to reach mid-log phase (OD600 0.3 to 0.5) and transferred into media that contained 0.2% arabinose to induce SpyCas9. At transfer, volumes were normalized by OD600 value to ensure that an equal number of cells were used (diluted to a final OD600 of 0.05 in the arabinose-containing medium). This second medium either contained or lacked 100 ng/ml doxycycline to control expression of *acrIIA22* or *gfp*, as with the initial media. Throughout this experiment, we included a control strain that lacked pZE21_tetR and only expressed SpyCas9. Kanamycin and doxycycline were omitted from its growth media. For this control strain, we also toggled the addition of arabinose in the second growth medium to ensure that positive and negative controls for SpyCas9 expression were included in our experiment. After 3 hours and 6 hours of SpyCas9 induction, OD600 readings were again taken and these values used to harvest an equal number of cells per sample (at 3 hours, OD600 values were between 0.76 and 0.93 and 0.75 ml to 0.9 ml volumes harvested; at 6 hours, 0.4 ml was uniformly harvested as all absorbance readings were approximately 1.6).

All samples were centrifuged at 4,100*g* to pellet cells, resuspended in 100 μl of denaturing lysis buffer (12.5 mM Tris-HCl (pH 6.8); 4% SDS), and passed through a 25-gauge needle several times to disrupt the lysate. Samples were then boiled at 100 ˚C for 10 minutes, spun at 13,000 rpm at 4 ˚C for 15 minutes and the supernatants removed and frozen at −20 ˚C. The next day, 12 μl of lysate was mixed with 4 μl of 4× sample buffer (200 mM Tris-HCl, 8% SDS, 40% glycerol, 200 mM DTT, and 0.05% bromophenol blue) and boiled at 100 ˚C for 10

minutes. Then, 10 μl sample was loaded onto a BioRad Mini-Protean "any KD Stain Free TGX" gel (cat. #4569035) and run at 150 V for 62 minutes. To verify that equivalent amounts of each sample were run, gels were visualized on a BioRad ChemiDoc for total protein content. Protein was then transferred to a 0.2-μM nitrocellulose membrane using the Bio-Rad Trans-Blot Turbo system (25 V, 1.3 A for 10 minutes). We then washed membranes in PBS/0.1% Triton-X before incubating them with a mixture of the following 2 primary antibodies, diluted in in LI-COR Odyssey Blocking Solution (cat. #927–40000): (i) monoclonal anti-SpyCas9, Diagenode cat. #C15200229-50, diluted 1:5,000; (ii) polyclonal anti-GAPDH, GeneTex cat. #GTX100118, diluted 1:5,000. The GAPDH antibody served as a loading control and a second check to ensure equal protein levels were run. Membranes were left shaking overnight at 4 ˚C, protected from light. Then, membranes were washed 4 times in PBS/0.1% Triton-X (10-minute washes) before they were incubated for 30 minutes at room temperature with a mixture of secondary antibodies conjugated to infrared dyes. Both antibodies were diluted 1:15,000 in LI-COR Odyssey Blocking Solution. To detect SpyCas9, the following secondary antibody was used: IR800 donkey, anti-mouse IgG, LI-COR cat# 926–32212. To detect GAPDH, IR680 goat, anti-rabbit IgG, LI-COR cat# 926–68071 was used. Blots were imaged on a LI-COR Odyssey CLx after 3 additional washes.

## Phage plaquing assay

We grew overnight cultures of *E. coli* Turbo expressing pSpyCa9_crMu and pZE21_tetR encoding a gene of interest (S4 and S5 Tables) at 37 ˚C in LB spec50 + kan50 + 0.5 mM IPTG. Genes of interest were either *acrIIA4*, *gfp*, or *acrIIA22*. The pSpyCas9 construct targeted phage Mu and was previously demonstrated to confer strong anti-phage immunity in this system [12]. A control strain expressing pZE21-tetR-*gfp* and SpyCas9_crNT (which encoded a crRNA that does not target phage Mu) was grown similarly. The next morning, all cultures were diluted 50-fold into LB spec50 + kan50 + 0.5 mM IPTG + 5 mM MgCl2 and grown at 37 ˚C for 3 hours. Then, doxycycline was added to a final concentration of 100 ng/ml to induce the gene of interest. Two hours later, SpyCas9 was induced by adding a final concentration of 0.2% w/v arabinose. Two hours after that, cultures were used in soft-agar overlays on one of 2 media types, discordant for arabinose, to either maintain SpyCas9 expression or let it fade as arabinose was diluted in top agar and consumed by the host bacteria (per S2 Fig). Top and bottom agar media were made with LB spec50 + kan50 + 0.5 mM IPTG + 5 mM MgCl2. In cases where SpyCas9 expression was maintained, arabinose was also added at a final concentration of 0.02% to both agar types. Top agar was made using 0.5% Difco agar and bottom agar used a 1% agar concentration. For the plaquing assay, 100 μl of bacterial culture was mixed with 3 ml of top agar, allowed to solidify, and 10-fold serial dilutions of phage Mu spotted on top using 2.5 μl droplets. After the droplets dried, plates were overturned and incubated at 37 ˚C overnight before plaques were imaged the following day.

## Identification of AcrIIA22 homologs and hypervariable genomic islands

We searched for AcrIIA22 homologs in 3 databases: NCBI nr, IMG/VR, and a set of assembled contigs from 9,428 diverse human microbiome samples [18]. Accession numbers for the NCBI homologs are indicated on the phylogenetic tree in Fig 3A. We retrieved AcrIIA22 homologs via 5 rounds of an iterative PSI-BLAST search against NCBI nr performed on October 2, 2017. In each round of searching, at least 90% of the query protein (the original AcrIIA22 hit) was covered, 88% of the subject protein was covered, and the minimum amino acid identity of an alignment was 23% (minimum 47% positive residues; e-value ≤ 0.001). Only one unique AcrIIA22 homolog was identified in IMG/VR (from several different phage genomes) via a

blastp search against the July 2018 IMG/VR proteins database (using default parameters). This homolog was also found in other databases, and its amino acid sequence is identical to that of AcrIIA22b (Fig 3A).

Most unique AcrIIA22 homologs were identified in the assembly data of over 9,400 human microbiomes performed by Pasolli and colleagues [18]. These data are grouped into multiple datasets: (i) the raw assembly data; and (ii) a set of unique species genome bins (SGBs), which were generated by first assigning species-level phylogenetic labels to each assembly and then selecting one representative genome assembly per species. We identified AcrIIA22 homologs using several queries against both databases. First, we performed a tblastn search against the SGB database using the AcrIIA22 sequence as a query, retrieving 141 hits from 137 contigs. A manual inspection of the genome neighborhoods for these hits revealed that most homologs originated from a short, hypervariable genomic island; some homologs were encoded by prophages. No phage-finding software was used to identify prophages; they were apparent from a manual inspection of the gene annotations that neighbored *acrIIA22* homologs (see the section entitled "Annotation and phylogenetic assignment of metagenomic assemblies" for details).

To find additional examples of AcrIIA22 homologs and of these genomic islands, we then queried the full raw assembly dataset. To do so without biasing for *acrIIA22*-encoding sequences, we used the *purF* gene that flanked *acrIIA22*-encoding genomic islands as our initial query sequence. Specifically, we used the *purF* gene from contig number 1 in S3 Table; its sequence is also in S5 Table. To consider only the recent evolutionary history of this locus, we required all hits have ≥98% nucleotide identity and required all hits to be larger than 15 kilobases in length to ensure sufficient syntenic information. From these contigs, we further filtered for those that had ≥98% nucleotide identity to *radC*, the gene that flanked the other end of *acrIIA22*-encoding genomic islands. Again, we used the variant from contig number 1 in S3 Table; its sequence is also in S5 Table. In total, this search yielded 258 contig sequences; nucleotide sequences and annotations for these contigs are provided in S6 Data. We then searched for *acrIIA22* homologs in these sequences using tblastn, again observing them in genomic islands and prophage genomes (which were assembled as part of the 258 contigs). In total, this search revealed 320 *acrIIA22* homologs from 258 contigs. The 258 genomic islands from these sequences were retrieved manually by extracting all nucleotides between the *purF* and *radC* genes. These extracted sequences were then clustered at 100% nucleotide identity with the sequence analysis suite Geneious Prime 2020 v1.1 to identify 128 unique genomic islands.

Altogether, our 2 searches yielded 461 AcrIIA22 sequences from these metagenomic databases that spanned 410 contig sequences. The 461 AcrIIA22 homologs broke down into 2 groups: 410 clustered with genomic island-like sequences, whereas 51 clustered with prophage-like homologs. In nature, the relative prevalence of AcrIIA22 in genomic islands or prophages may not be accurately reflected by these numbers because we never directly searched for prophage-encoded homologs. We then combined these 461 AcrIIA22 sequences with those from NCBI and IMG/VR and clustered the group on 100% amino acid identity to reveal 30 unique proteins. To achieve this, we used the software cd-hit [49] with the following parameters: -d 0 -g 1 -aS 1.0 -c 1.0. These 30 sequences were numbered to match one of their parent contigs (as indicated in S3 Table) and used to create the phylogenetic tree depicted in Fig 3A. For AcrIIA22 homologs found outside NCBI, the nucleotide sequences and annotations of their parent contigs can be found in S2 and S3 Data. For NCBI sequences, accession numbers are shown in Fig 3A. The gene sequences used in functional assays (Fig 3B) have been reprinted in S5 Table for convenience.

## Annotation and phylogenetic assignment of metagenomic assemblies

Contig sequences from IMG/VR, the Pasolli metagenomic assemblies, and some NCBI entries lacked annotations, making it difficult to make inferences about *acrIIA22's* genomic neighborhood. To facilitate these insights, we annotated all contigs as follows. We used the gene-finder MetaGeneMark [50] to predict ORFs using default parameters. We then used their amino acid sequences in a profile HMM search with HMMER3 [51] against TIGRFAM [52] and Pfam [53] profile HMM databases. The highest scoring profile was used to annotate each ORF. We annotated these contigs to facilitate genomic neighborhood analyses for *acrIIA22*; these are not intended to provide highly accurate functional predictions of their genes. Thus, we erred on the side of promiscuously assigning gene function; our annotations should therefore be treated with appropriate caution. A visual inspection of these annotated contigs made apparent several examples of *acrIIA22*-encoding prophages (we noticed 35- to 40-kilobase insertions in some contigs that were otherwise nearly identical to those without prophages). We were confident that these insertions were prophages because they contained mostly colinear genes with key phage functions annotated. As a simple means to sample this phage diversity, we manually extracted 9 examples of these prophage sequences (their raw sequences and annotated genomes can be found in S4 and S5 Data). Annotations were imported into the sequence analysis suite Geneious Prime 2020 v1.1 for manual inspection of genome neighborhoods.

We used the genome taxonomy database (GTDB) convention for all sequences discussed in this manuscript [54]. In part, this was because all *acrIIA22* genomes are found in clostridial genomes, which are notoriously polyphyletic in NCBI taxonomies (for instance, species in the NCBI genus *Clostridium* appear in 121 GTDB genera and 29 GTDB families) [55]. All SGBs that we retrieved from the Pasolli assemblies were assigned taxonomy as part of that work and were called *Clostridium* sp. CAG-217. Similarly, NCBI assemblies that encoded the most closely *acrIIA22* homologs to our original hit were assigned to the GTDB genus CAG-217 [54,55]. The raw assembly data from the Pasolli database were not assigned a taxonomic label but were nearly identical in nucleotide composition to the CAG-217 contigs (Fig 2 and S4 Fig; S2 and S3 Data). Therefore, we also refer to these sequences as originating in CAG-217 genomes but take care to indicate which assemblies have been assigned a rigorous taxonomy and which ones for which taxonomy has been inferred in this fashion (S3 Table).

## Comparing genes in genomic islands to phage genomes

We first examined the annotated genes within each of the 128 unique genomic islands. Manual inspection revealed 54 unique gene arrangements that differed in gene content and orientation. We then selected one representative from each arrangement and extracted amino acid sequences from each encoded gene ($n$ = 506). Next, we collapsed these 506 proteins into orthologous groups by clustering at 65% amino acid using cd-hit with the following parameters: -d 0 -g 1 -aS 0.95 -c 0.65. These cluster counts were used to generate the histogram depicted in S4C Fig. To determine which protein families may also be phage-encoded, we queried the longest representative from each cluster with at least 2 sequences against the database of 9 CAG-217 phages described in the section entitled "Annotation and phylogenetic assignment of metagenomic assemblies." We used tblastn with default parameters to perform this search, which revealed that some proteins in the CAG-217 genomic islands have homologs in prophage genomes that are out of frame with respect to the MetaGeneMark annotations depicted in S4A Fig.

## Phylogenetic tree of AcrIIA22 homologs

The 30 unique AcrIIA22 homologs we retrieved were used to create the phylogeny depicted in Fig 3A. These sequences were aligned using the sequence alignment tool in the sequence analysis suite Geneious Prime 2020 v1.1. This alignment is provided as S7 Data. From this alignment, the phylogenetic tree in Fig 3A was generated using PhyML with the LG substitution model [56] and 100 bootstraps. Coloration and tip annotations were then added in Adobe Illustrator.

## Identification of CRISPR-Cas systems and Acrs in CAG-217 assemblies

To determine the type and distribution of CRISPR-Cas systems and Acrs in CAG-217 genomes, we downloaded all assembly data for the 779 SGBs assigned to CAG-217 in Pasolli and colleagues [18] (bin 4303). We then predicted CRISPR-Cas systems for all 779 assemblies in bulk using the command line version of the CRISPR-Cas prediction suite, cctyper [57]. Specifically, we used version 1.2.1 of cctyper with the following options:—prodigal meta—keep_tmp. To identify type II-A Acrs, we first downloaded representative sequences for each of the 21 experimentally confirmed type II-A Acrs from the unified resource for tracking Acrs [58]. We then used tblastn to query these proteins against the 779 CAG-217 genome bins and considered any hit with e-value better than 0.001 (which included all hits with >30% identity across 50% of the query). To check if these Acrs were present in *acrIIA22*-encoding phages, we performed an identical tblastn search, but this time using the set of 9 *acrIIA22*-encoding prophages as a database.

## Recombinant protein overexpression and purification

The AcrIIA22 protein and its mutants were codon optimized for *E. coli* (Genscript or SynBio Technologies), and the gene constructs were cloned into the pET15HE or pET15b plasmid [12] to contain an N-terminal, thrombin-cleavable 6XHistidine (His6) tag. These plasmids differ by only a few bases just upstream of the N-terminal thrombin tag. For purified, twin-strep tagged proteins, constructs were cloned into a modified pET15b that lacks the N-terminal tag but instead has a C-terminal twin-strep tag (S4 Table). Constructs were transformed and overexpressed in BL21 (DE3) RIL or BL21 (DE3) pLysS *E. coli* cells. A 10-mL overnight culture (grown in LB + 100 µg/mL ampicillin) was diluted 100-fold into the same media and grown at 37 ˚C with shaking to an OD600 of 0.8 for His6-tagged constructs and 0.3 for twin-strep-tagged constructs. Expression was then induced with 0.5 mM IPTG. For His6-tagged constructs, the culture was shaken for an additional 3 hours at 37 ˚C; twin-strep-tagged constructs were induced at 16 ˚C for 22 hours. Cells were harvested by centrifugation, and the pellet stored at −20 ˚C.

Cell pellets for His6-tagged constructs were resuspended in 25 mM Tris (pH 7.5), 300 mM NaCl, 20 mM imidazole (lysis buffer); twin-strep tagged constructs were resuspended in Tris 100 nM (pH 8.0), 150 mM NaCl, 1 mM EDTA. Cells were lysed by sonication on ice. The lysate was centrifuged in an SS34 rotor at 18,000 rpm for 25 minutes, followed by filtering through a 5-µm syringe filter for the His6-tagged constructs and a 0.45-µM syringe filter for the twin-strep-tagged constructs.

To purify His6-tagged constructs, the clarified lysate was bound using the batch method to Ni-NTA agarose resin (Qiagen, Germantown, MD, USA) at 4 ˚C for 1 hour. The resin was transferred to a gravity column (Bio-Rad, Hercules, CA, USA), washed with >50 column volumes of lysis buffer, and eluted with 25 mM Tris (pH 7.5), 300 mM NaCl, 200 mM imidazole. The protein was diluted with 2 column volumes of 25 mM Tris (pH 7.5) and purified on a HiTrapQ column (Cytivia life sciences, Marlborough, MA, USA) using a 20-mL gradient from

150 mM to 1 M NaCl in 25 mM Tris (pH 7.5). Peak fractions were pooled, concentrated, and buffer exchanged into 200 mM NaCl, 25 mM Tris (pH 7.5) using an Amicon Ultra centrifugal filter with a 3,000 molecular weight cutoff (MilliporeSigma (Burlington, MA, USA), UFC900324), then cleaved in an overnight 4 °C incubation with biotinylated thrombin (MilliporeSigma, Burlington, MA, USA). Streptavidin agarose slurry (Novagen, Birmingham, UK) was incubated with cleaved protein at 4 °C for 30 minutes to remove thrombin. The sample was then passed through a 0.22-μm centrifugal filter and loaded onto a HiLoad 16/60 Superdex 200 prep grade size exclusion column (MilliporeSigma, Burlington, MA, USA) equilibrated in 25 mM Tris (pH 7.5), 200 mM NaCl. The peak fractions were pooled, concentrated, and confirmed for purity by SDS-PAGE before use in most assays. Fig 4B depicts SEC data generated for thrombin-cleaved AcrIIA22 variants generated using a Superdex75 16/60 (GE HealthCare) column with 25 mM Tris (pH 7.5), 200 mM NaCl. To correlate nicking activity with protein content across fractions (S10B Fig), we collected 13 fractions that span the entire elution peak as well as fractions without AcrIIA22 protein. The protein gel shown in S10A and S10B Fig was loaded with 5 μl of each concentrated fraction.

For 2 additional proteins, we also performed similar Ni-NTA–based purifications of His6-tagged constructs, with small deviations from the protocol described in the preceding paragraph. Recombinant AcrIIA4 was purified similarly to other His6-tagged Acr proteins but with the following deviations, as previously described [12]. IPTG was used at 0.2 mM, and cells were harvested after 18 hours of induction at 18 °C. Thrombin cleavage also occurred at 18 °C. This untagged version was used to help generate S6 Fig. Peak fractions for all proteins were pooled, concentrated, flash frozen as single-use aliquots in liquid nitrogen, and stored at −80 °C. SpyCas9 was expressed in *E. coli* from plasmid pMJ806 (addgene #39312) to contain a TEV-cleavable N-terminal 6XHis-MBP tag and was purified as described previously [12] with sequential steps of purification consisting of Ni-NTA affinity chromatography, TEV cleavage, Heparin HiTrap chromatography, and SEC. The protein was stored in a buffer consisting of 200 mM NaCl, 25 mM Tris (pH 7.5), 5% glycerol, and 2 mM DTT. Again, peak fractions were pooled, concentrated, and flash frozen as single-use aliquots.

We also purified AcrIIA22 and AcrIIA4 constructs with a C-terminal twin-strep tag. The protein was expressed and lysed as described above and purified according to the manufacturer's guidelines (IBA Life Sciences). Clarified lysates were passed over Strep-Tactin-Sepharose resin using a gravity filtration column. The flow through was passed over the resin a second time. The column was washed with a minimum of 20 column volumes of buffer W, followed by elution in buffer E (150 mM NaCl, 100 mM Tris (pH 8.0), 1 mM EDTA, 2.5 mM desthiobiotin). The eluted protein was purified over a HiTrapQ column (GE Healthcare) using a 40-mL gradient from 150 mM to 0.5 M NaCl in 25 mM Tris (pH 7.5). Peak fractions were pooled and then purified again via SEC with a Biorad Enrich SEC650 10 × 300 mm column in 150 mM NaCl, 25 mM Tris (pH 7.5). These elution data are shown for AcrIIA22 and its variants in Fig 6B. Fractions were collected across the elution peak and confirmed for purity via silver stain (S10E Fig), per manufacturer's recommendations (Thermo Fisher Cat. No. 24612). For these proteins, we chose fraction number 4 to carry forward, as it eluted at approximately 4 times the monomer's molecular weight, consistent with our proposed tetramer, which is depicted in Fig 4C. Protein was then concentrated and flash frozen as single-use aliquots for later use.

## X-ray crystallography and structural analyses

An AcrIIA22 crystal was grown using 14 mg/mL protein via the hanging drop method using 200 mM ammonium nitrate, 40% (+/−)-2-methyl-2,4-pentanediol (MPD, Hampton Research, Aliso Viejo, CA, USA), 10 mM MgCl2 as a mother liquor. Diffraction data were collected at

the Argonne National Laboratory Structural Biology Center synchrotron facility (Beamline 19BM). Data were processed with HKL2000 in space group P4332, then built and refined using COOT [59] and PHENIX [60]. The completed 2.80 Å structure was submitted to the Protein Data Bank with PDB Code 7JTA. The detailed PDB validation report is provided (S8 Data). We submitted this finished coordinate file to the PDBe PISA server (Protein Data Bank Europe, Protein Interfaces, Surfaces and Assemblies; http://pdbe.org/pisa/), which uses free energy and interface contacts to calculate likely multimeric assemblies [27]. The server calculated tetrameric, dimeric, and monomeric structures to be thermodynamically stable in solution. The tetrameric assembly matches the molecular weight expected from the size exclusion column elution peak and is the most likely quaternary structure as calculated by the PISA server. The tetramer gains −41.8 kcal/mol free energy by solvation when formed and requires an external driving force of 3.1 kcal/mol to disassemble it according to PISA ΔG calculations.

## sgRNA generation

The sgRNA for use in in vitro experiments was generated as described previously [12]. We made the dsDNA template via one round of thermal cycling (98 ˚C for 90 seconds, 55 ˚C for 15 seconds, 72 ˚C for 60 seconds) in 50 μl reactions. We used the Phusion PCR polymerase mix (NEB, Ipswich, MA, USA) containing 25 pmol each of the following 2 oligo sequences; the sequence that binds the protospacer on our pIDTsmart target vector is underlined:

1. GAAATTAATACGACTCACTATAGG<u>TAATGAAATAAGATCACTAC</u>GTTTTAGAGC TAGAAATAGCAAGTTAAAATAAGGCTAGTCCG

2. AAAAAAGCACCGACTCGGTGCCACTTTTTCAAGTTGATAACGGACTAGCCTT ATTTTAACTTGC.

   The dsDNA templates were then purified using an Oligo Clean and Concentrator Kit (Zymo Research, Irivine, CA, USA) before quantification via the Nanodrop. sgRNA was transcribed from this dsDNA template by T7 RNA polymerase using Megashortscript Kit (Thermo Fisher #AM1354). Reactions were then treated with DNAse, extracted via phenol-chloroform addition and then chloroform addition, ethanol precipitated, resuspended in RNase-free water, quantified by Nanodrop, analyzed for quality on 15% acrylamide/TBE/UREA gels, and frozen at −20 ˚C.

## Pulldown assay using twin-strep-tagged AcrIIA22 and AcrIIA4

The same buffer, consisting of 200 mM NaCl, 25 mM Tris (pH 7.5), was used for pulldowns and to dilute proteins. As a precursor to these assays, 130 pmol SpyCas9 and sgRNA were incubated together at room temperature for 15 minutes where indicated. SpyCas9, with or without precomplexed sgRNA, was then incubated with 230 pmol AcrIIA4 or 320 pmol AcrIIA22 for 25 minutes at room temperature. Subsequently, 50 μl of a 10% slurry of Strep-Tactin Resin (IBA Lifesciences #2-1201-002), which was preequilibrated in binding buffer, was added to the binding reactions and incubated at 4 ˚C on a nutator for 45 minutes. Thereafter, all incubations and washes were carried out at 4 ˚C or on ice. Four total washes of this resin were performed, which included one tube transfer. Washes proceeded via centrifugation at 2,000 rpm for 1 minute, aspiration of the supernatant with a 25-gauge needle, and resuspension of the beads in 100 μl binding buffer. Strep-tagged proteins were eluted via suspension in 40 μl of 1× BXT buffer (100 mM Tris-Cl, 150 mM NaCl, 1 mM EDTA, 50 mM Biotin (pH 8.0)) and incubated for 15 minutes at room temperature. After centrifugation, 30 μl of supernatant was removed and mixed with 4× reducing sample buffer (Thermo Fisher, Waltham,

MA, USA). Proteins were then separated by SDS PAGE on BOLT 4% to 12% gels in MES buffer (InvitrogenThermo Fisher, Waltham, MA, USA) and visualized by Coomassie staining.

## SpyCas9 linear DNA cleavage assay

All SpyCas9 cleavage reactions using linear DNA were performed in cleavage buffer [61] (20 mM Tris HCl (pH7.5), 5% glycerol, 100 mM KCl, 5 mM MgCl2, 1 mM DTT). In preparation for these reactions, all proteins were diluted in 30 mM NaCl/25 mM Tris (pH 7.4)/2.7 mM KCl, whereas all DNA and sgRNA reagents were diluted in nuclease-free water. Where indicated, SpyCas9 (0.36 μM) was incubated with sgRNA (0.36 μM) for 10 minutes at room temperature. Before use, sgRNA was melted at 95 ˚C for 5 minutes and then slowly cooled at 0.1 ˚C/s to promote proper folding. SpyCas9 (either precomplexed with sgRNA or not, as indicated in S7 Fig) was then incubated for 10 minutes at room temperature with AcrIIA4 (2.9 μM) or AcrIIA22 at each of the following concentrations: [23.2, 11.6, 5.8, and 2.9 μM]. As substrate, the plasmid pIDTsmart was linearized by restriction digest and used at a final concentration of 3.6 nM. The reaction was initiated by the addition of this DNA substrate either in isolation or in combination with sgRNA (0.36 μM) as indicated in S7 Fig. Reactions were immediately moved to a 37 ˚C incubator, and the reaction stopped after 15 minutes via the addition of 0.2% SDS/100 mM EDTA and incubation at 75 ˚C for 5 minutes. Samples were then run on a 1.5% TAE agarose gel at 120 V for 40 minutes. Densitometry was used to calculate the proportion of DNA cleaved by SpyCas9; band intensities were quantified using the BioRad ImageLab software v5.0.

## In vivo assay to assess impact of AcrIIA22 on plasmid topology

In all experiments, cultures were first grown overnight at 37 ˚C with shaking at 220 rpm in LB with 0.5 mM IPTG and, if included, spectinomycin at 50 μg/mL, and kanamycin at 50 μg/mL. For each sample with a SpyCas9-expressing plasmid (e.g., Fig 7A), overnight cultures were grown with spectinomycin and kanamycin and diluted 1:50 into LB with 0.5 mM IPTG, spectinomycin (at 50 μg/mL), and, where indicated, doxycycline (at 100 ng/mL, to induce *acr*s). Cultures were grown at 37 ˚C with shaking at 220 rpm. If required, 0.2% (L)-arabinose was added after 2 hours of growth to induce *spyCas9* expression. The next morning, cultures were centrifuged at 4,100*g* and plasmids purified using a miniprep kit (Qiagen). We measured the concentration of dsDNA in each miniprep using the Qubit-4 fluorometer and the associated dsDNA high-sensitivity assay kit (Invitrogen). For each sample with a SpyCas9-expressing plasmid, 150 ng of DNA was digested with the restriction enzyme HincII (NEB) per manufacturer's recommendations, except that digests were incubated overnight before being stopped by heating at 65 ˚C for 20 minutes. This restriction enzyme will cut once, only in the SpyCas9 plasmid, to linearize it. This allowed us to visualize the SpyCas9 plasmid as a single band, which allowed us to identify bands from *acrIIA22*-encoding undigested plasmids more easily. It also served as an internal control for plasmid DNA that is unaffected by SpyCas9 targeting or AcrIIA22 expression (S2 Fig). Following restriction digest, 30 ng of sample was analyzed via gel electrophoresis using a 0.7% TAE-agarose gel run at 120 V for 30 minutes.

In samples that lacked a SpyCas9-expressing plasmid (e.g., Fig 5A), overnight cultures were grown with kanamycin and diluted into LB. Where required, 0.5 mM IPTG and doxycycline at 100 ng/mL were added to induce the gene of interest. The next morning, cultures were centrifuged at 4,100*g* and plasmids purified using a miniprep kit (Qiagen). The concentration of dsDNA in each miniprep was measured using the Qubit-4 fluorometer and the associated dsDNA high-sensitivity assay kit (Invitrogen). Then, 30 ng of purified plasmid was directly analyzed by gel electrophoresis using a 0.7% TAE-agarose gel run at 120 V for 30 minutes.

## In vitro AcrIIA22 plasmid nicking assay

Except for the divalent cation experiment, all reactions were performed using NEB buffer 3.1 (100 mM NaCl, 50 mM Tris-HCl (pH 7.9), 10 mM MgCl2, 100 µg/mL BSA). To determine cation preference, the same reaction buffer was re-created, but MgCl2 was omitted. All proteins were diluted in 130 mM NaCl, 25 mM Tris (pH 7.4), 2.7 mM KCl. DNA was diluted in nuclease-free water. In the cation preference experiment, 60 µM His6-AcrIIA22 and 6 nM of purified pIDTsmart plasmid DNA were used. All other reactions were set up with AcrIIA22 constructs and concentrations indicated in figure panels and captions. In the cation preference experiment (S11A Fig), reactions were started by adding 10 mM of the indicated cation. All other reactions were initiated via the addition of 2 nM pIDTsmart plasmid DNA. In these cases, reactions were immediately transferred to a 37 ˚C incubator. At 0.5-, 1-, 2-, 4-, 6-, or 20-hour time points, a subset of the reaction was removed and run on a 1.5% TAE agarose gel at 120 V for 30 minutes. For the fractionation experiment depicted in S10B Fig, 5 µl of each concentrated fraction was used in a 15-µl reaction volume, and the reaction was incubated for 24 hours at 37 ˚C. For the cation preference experiment, only the 2-hour time point was considered, and the reaction was stopped via the addition of NEB loading buffer and 100 mM EDTA. In this case, DNA was visualized on a 1% TBE gel run for 60 minutes at 110 V. Densitometry was used to calculate the proportion of DNA in each topological form via band intensities quantified using the BioRad ImageLab software v5.0.

## SpyCas9 cleavage kinetics assay

All cleavage reactions were performed in the cleavage buffer [61] containing 20 mM Tris HCl (pH7.5), 5% glycerol, 100 mM KCl, 5 mM MgCl2, 1 mM DTT. In preparation for these reactions, all proteins were diluted in 30 mM NaCl/25 mM Tris (pH 7.4)/2.7 mM KCl, whereas all DNA and sgRNA reagents were diluted in nuclease-free water.

Purified pIDTsmart plasmid was pretreated with either AcrIIA22, the nickase Nb.BssSI (NEB), or no enzyme. For the AcrIIA22 pretreatment, 3.1 µg of plasmid was incubated with 230 µM AcrIIA22 and the plasmid nicked as described previously. Plasmid nicking with Nb. BssSI was carried out via manufacturer's recommendations (NEB). Both reactions were incubated at 37 ˚C for 2 hours. To isolate the nicked plasmid, samples were then run on a 1.5% agarose gel for 2 hours and the open-circle form of the plasmid was excised and purified using the Zymo Research Gel DNA Recovery Kit. Untreated plasmid was also purified via gel extraction. Plasmid yield was quantified using a Nanodrop.

To determine SpyCas9's substrate preference, we incubated each pretreated plasmid substrate with SpyCas9 and assayed for the appearance of a linearized plasmid as indication of SpyCas9 digestion. In all cases, SpyCas9 was used at a final concentration of 0.32 µM. All reaction components except dsDNA were added on ice, following which SpyCas9 was complexed with equimolar levels of its sgRNA for 10 minutes at room temperature. Before addition to the reaction, sgRNA was melted at 95 ˚C for 5 minutes and then slowly cooled at 0.1 ˚C/s to promote proper folding. To begin the reaction, DNA substrate was added to the reaction mix at a final concentration of 2 nM, and the samples moved immediately to 37 ˚C. At each time point, a subset of the reaction was removed, and digestion stopped with 0.2% SDS/100 mM EDTA and by incubating at 75 ˚C for 5 minutes. Samples were run on a 1.5% TAE gel at 120 V for 40 minutes, and densitometry was used to calculate the proportion of DNA in each topological form via band intensities quantified with the BioRad ImageLab software v5.0.

## Supporting information

**S1 Fig. *Orf_1* (*acrIIA22*) confers mild toxicity in *E. coli*.** Growth rates with *orf_1* induction (green) are 7% lower than those without *orf_1* induction (orange). The CFU data shown in Fig 1C were generated from the same experiment depicted here (samples were removed after 6 hours of growth to determine these CFU counts). Thus, these data demonstrate that anti-Spy-Cas9 activity occurs under conditions with mild *orf_1* toxicity. Growth curves are shown for samples without SpyCas9 induction to ensure that *orf_1* toxicity is not mitigated due to elimination of its plasmid. Points indicate averages from 3 replicates. Standard deviations at each time point are so small that the error bars do not exceed the bounds of the data point. The individual numerical values that underlie the summary data in this figure may be found in S1 Data. CFU, colony-forming unit; SpyCas9, *Streptococcus pyogenes* Cas9.
(PDF)

**S2 Fig. *Orf_1 (acrIIA22*) does not impact SpyCas9 expression.** (**A**) A schematic description of the experimental design shown in panel (**B**) is presented. If ORF_1 prevented transcription from pCas9 or altered its copy number, we would expect expression of the *orf_1* gene to deplete the level of green fluorescence observed from a construct that replaces the *spycas9* gene with *gfp*. (**B**) Fluorescence measurements for the experiment depicted in panel A show that ORF_1 does not impact GFP expression throughout an *E. coli* growth curve. Points indicate averages from 3 replicates; error bars indicate standard deviation. A western blot shows no depletion of SpyCas9 expression as a function of ORF_1 or GFP expression in growing *E. coli* cultures at 3 hours (**C**) or 6 hours **(D)**. As an internal control, GAPDH expression was also detected. The individual numerical values and original images for the data presented in this figure may be found in S1 Data and S1 Raw Images, respectively. SpyCas9, *Streptococcus pyogenes* Cas9.
(PDF)

**S3 Fig. AcrIIA22 only modestly protects Mu phages against SpyCas9.** Mu phage fitness was measured by plaquing on *E. coli* in the presence of *gfp*, *acrIIA22*, or *acrIIA4* via serial 10-fold dilutions. Bacterial clearing (black) occurs when phage Mu overcomes SpyCas9 immunity and lyses *E. coli*. In (**A**) and in (**B**), SpyCas9 with a Mu-targeting crRNA confers substantial protection against phage Mu relative to an n.t. control, in both conditions tested. These conditions are depicted at left, with the only difference being whether SpyCas9 was only expressed in liquid growth prior to phage infection (panel A) or expressed both in liquid media and in solid media throughout infection (panel B). When expressed from a second plasmid, the positive control *acrIIA4* significantly enhances Mu fitness by inhibiting SpyCas9 in all conditions *in trans*. Though *acrIIA22* confers protection against SpyCas9 compared to *gfp* (negative control), this effect is milder than with *acrIIA4* and dependent on SpyCas9 expression. crRNA, CRISPR RNA; n.t., nontargeting; SpyCas9, *Streptococcus pyogenes* Cas9.
(PDF)

**S4 Fig. *AcrIIA22* homologs are found in hypervariable regions of prophage and bacterial genomes in the CAG-217clostridial genus.** (**A**) Homologs of *acrIIA22* are depicted in 3 related prophage genomes, integrated at 3 different genomic loci, revealed by a comparison of prophage-bearing contigs (#57, #56, #37) relative to unintegrated contigs (#55, #58, #17, respectively), which are otherwise nearly identical. Prophage genes are colored by functional category, according to the legend at the left of panel A. Genes immediately adjacent to *acrIIA22* (solid boxes) vary across phages, despite strong relatedness across much of the prophage genomes. Bacterial genes are colored gray, except for contig #17, which is also depicted in panel B, below. (**B**) Homologs of *acrIIA22* are depicted in diverse genomic islands,

including Contig #1, whose sequence includes a portion that is identical to the original meta-genomic contig we recovered (F01A_4). All *acrIIA22* homologs in these loci are closely related but differ in their adjacent genes, which often have homologs in the prophages depicted in panel A (dashed boxes). Bacterial genomic regions flanking these hypervariable islands are nearly identical to one another and to prophage integration locus #3, as shown by homology to contig #17 from panel A. Contigs are numbered to indicate their descriptions in S3 Table, which contains their metadata, taxonomy, and sequence retrieval information. All sequences and annotations can also be found in S2 and S3 Data. (**C**) We tabulate the prevalence of various protein families (clustered at 65% amino acid identity) in a set of 54 unique genomic islands. Each of these islands is flanked by the conserved genes *purF* and *radC* but contains a different arrangement of encoded genes. Domain-level annotations are indicated below each protein family. Gene symbols above each protein family are colored and lettered to indicate their counterparts or homologs in panels A and B. The phage capsid icon indicates sequences with homologs in prophage genomes. (**D**) An evolutionary model for the origin of the *acrIIA22*-encoding hypervariable genomic islands depicted in panel B is shown. This panel is reprinted from Fig 2C for continuity. We propose that *acrIIA22* moved via a phage insertion into a bacterial genomic locus, remained following an incomplete prophage excision event, and its neighboring genes subsequently diversified via horizontal exchange with additional phage genomes. The individual numerical values that underlie the summary data in this figure may be found in S1 Data. ORF, open reading frame; unk; unknown function.
(PDF)

**S5 Fig. Genomic proximity of *acrIIA22* homologs to other *acr* genes.** An *acrIIA22*-encoding prophage like the one depicted in Fig 2A and those in S4A Fig is shown. This prophage encodes for a homolog of the previously described SpyCas9 inhibitor *acrIIA17* within 1 kilobase of an *acrIIA22* homolog. Sequence relatedness between the depicted *acrIIA17* gene and the originally discovered acrIIA17 is shown [22]. Because phages often encode multiple *acr*s in the same locus, the colocalization of *acrIIA17* with *acrIIA22* is consistent with the latter gene functioning natively to inhibit CRISPR-Cas activity. Prophage genes are colored by functional category, per the legend and as in S4A Fig. Contigs are numbered to indicate their descriptions in S3 Table, which contains their metadata, taxonomy, and sequence retrieval information. All sequences and annotations can also be found in S2 and S3 Data. Acr, anti-CRISPR; SpyCas9, *Streptococcus pyogenes* Cas9.
(PDF)

**S6 Fig. AcrIIA22 does not strongly bind SpyCas9.** SpyCas9 and sgRNA were preincubated before mixing with a TS-tagged AcrIIA22 or AcrIIA4. SpyCas9 without sgRNA was also used. Strep-Tactin pulldowns on AcrIIA4 also pulled down SpyCas9 preincubated with sgRNA, as previously reported [12]. Similar pulldowns with AcrIIA22 indicate little to no interaction with SpyCas9, regardless of whether sgRNA was used. These images depict total protein content visualized by Coomassie stain. Reaction components are indicated below the gel image. Asterisks (\*) and dagger (†) symbols indicate AcrIIA4 and AcrIIA22 protein bands that run at slightly different positions than expected due to gel distortion. Original, uncropped versions of images depicted in figure may be found in the Supporting information file, S1 Raw Images. sgRNA, single-guide RNA; SpyCas9, *Streptococcus pyogenes* Cas9; TS, twin-strep.
(PDF)

**S7 Fig. AcrIIA22 does not protect linear DNA from SpyCas9 cleavage.** (**A**) A schematic cartoon depicts the experiment in panel (B). SpyCas9 was preincubated with sgRNA targeting linear DNA. Then, Acr candidates were added. Subsequently, cleavage reactions were performed,

and the DNA products visualized by gel electrophoresis. (**B**) We show the products of the reactions described in panel A for the inhibitors AcrIIA22 and AcrIIA4. SpyCas9 activity is greatly inhibited by AcrIIA4 but unaffected by AcrIIA22, as indicated by the proportion of cleaved DNA product. Reaction components are depicted atop the gel image, with molar equivalents relative to SpyCas9 indicated. The percent of DNA substrate cleaved by SpyCas9 is quantified below each lane. (**C**) We perform a similar experiment as in panel A, except candidate Acrs were incubated with SpyCas9 before sgRNA addition. Reactions were begun via the simultaneous addition of sgRNA and linear dsDNA instead of just dsDNA. (**D**) The products of the reactions described in panel C for AcrIIA22 and AcrIIA4 inhibitors are shown. SpyCas9 activity is inhibited by AcrIIA4 but unaffected by AcrIIA22, as indicated by the proportion of cleaved DNA product. The data depicted in this figure are not directly comparable to those in Fig 7, due to methodological differences and because the preparations of SpyCas9 used in each experiment exhibited different activities. Original, uncropped versions of images depicted in figure may be found in the supporting information file, S1 Raw Images. Acr, anti-CRISPR; dsDNA, double-stranded DNA; SpyCas9, *Streptococcus pyogenes* Cas9; sgRNA, single-guide RNA.
(PDF)

**S8 Fig. AcrIIA22 resembles a PC4-like protein.** (**A**) We present a ribbon diagram of a proposed AcrIIA22 tetramer, which requires binding between antiparallel β-strands at the C-termini of AcrIIA22 monomers to form extended, concave β-sheets. This putative oligomerization interface is indicated by the regions highlighted in yellow. Each monomer in the proposed tetramer is labeled with lowercase Roman numerals (i-iv). (**B**) Space filling model of the tetrameric AcrIIA22 structure from panel A, with relative charge depicted, highlighting a groove (dashed line with arrowhead) that may accommodate nucleic acids (based on analogy to other PC4-like proteins). (**C**) AcrIIA22 monomers (i) and (ii) from the tetramer in panel A likely interact via a series of hydrophobic interactions, as indicated by the predominantly nonpolar sidechains colored in yellow. The boxed region highlights residue D14, which is important for nicking activity and plasmid protection against SpyCas9, and is enlarged in panel F. (**D**) In conventional PC4-like family proteins, such as the putative single-stranded DNA binding protein from phage T5 depicted in gray (PDB:4BG7), the same topology of outward facing, concave β-sheets are instead stabilized via interactions between opposing α-helices (depicted in opaque light blue). (**E**) An overlay of β-sheets from AcrIIA22 (blue, PDB:7JTA) and the phage T5 PC4-like protein (gray, PDB:4BG7) illustrates their similar topologies. (**F**) Two D14 residues in loop regions of AcrIIA22 monomers (i) and (ii) are in close proximity. These residues are important for nicking activity and may bind divalent cations in cells under physiological pH. (**G**) A close view of a putative salt bridge between R30 of monomers (i)/(ii) and the peptide backbone of the C-terminus of monomers (iv)/(iii), respectively. AcrIIA22 monomers are colored as described in panel A.
(PDF)

**S9 Fig. A 2-aa truncation mutant of AcrIIA22 is impaired for SpyCas9 inhibition and nicking activity.** (**A**) An in vivo plasmid protection assay. Asterisks depict statistically significant differences in plasmid retention under SpyCas9-inducing conditions with either wild-type AcrIIA22, a null mutant with an early stop codon, a 2-aa truncation, or a negative control *gfp* gene (adj. $p < 0.005$, Student $t$ test, $n = 3$). The truncation mutant retains mild but severely impaired activity, as it protects a plasmid from SpyCas9 more effectively than a null mutant ($p = 0.012$) or GFP control ($p = 0.015$). All $p$-values were corrected for multiple hypotheses using Bonferroni method. (**B**) The 2-aa truncation mutant is impaired for nicking in vitro, relative to wild-type AcrIIA22. In both cases, 25 μM of protein was used following Ni-NTA–

based purification of an N-terminal, His6-tagged construct. An asterisk (*) denotes significant differences between AcrIIA22-treated and untreated substrates (Student $t$ test, $p < 0.05$, $n = 3$). Standard deviations are indicated by dashed lines (in most cases, the data points obscure these error bars). The individual numerical values that underlie the summary data in this figure may be found in S1 Data. CFU, colony-forming unit; SpyCas9, *Streptococcus pyogenes* Cas9. (PDF)

**S10 Fig. AcrIIA22 nicks supercoiled plasmids.** (**A**) A Coomassie stain of an N-terminally His6-tagged AcrIIA22 construct shows no copurifying proteins. (**B**) The nicking activity for this protein preparation (bottom) correlates with the intensity of the Coomassie-stained protein band across purification fractions (top). In each lane, SC plasmid DNA represents the unnicked fraction, whereas OC and linear DNA have been nicked at least once. (**C**) This panel is a quantification of the experiment depicted in panel B across all 13 fractions collected. (**D**) His6-AcrIIA22 nicks SC plasmids in a time- and concentration-dependent manner. A decrease in the proportion of SC plasmid DNA indicates nicking activity, as depicted in Fig 5B. (**E**) A silver stain of a C-terminally twin-strep-tagged AcrIIA22 construct shows no copurifying proteins. Equal volumes of each protein fraction were loaded in each lane for all samples. Fraction 4 was concentrated and used for all in vitro experiments. (**F**) A C-terminal, but not N-terminal twin-strep tag, is compatible with AcrIIA22's ability to protect a target plasmid from SpyCas9 elimination in vivo. Statistically significant differences in plasmid retention between SpyCas9-inducing and noninducing conditions were determined via a Student $t$ test ($n = 3$); "**\***" indicates $p \leq 0.001$. All $p$-values were adjusted for multiple hypotheses using the Bonferroni correction. (**G**) The D14A mutation in AcrIIA22 impairs nicking activity. Over time, the wild-type AcrIIA22-twin-strep construct consistently converts a higher fraction of plasmid DNA from its SC form to an OC conformation than a D14A mutant. Control plasmids include a miniprepped sample and sample pretreated with the commercial nickase, Nb. BssSI. Reaction times are indicated to the right of each image. (**H**) AcrIIA22a (Fig 3B) is impaired for nicking activity relative to AcrIIA22. As in panel G, both constructs were purified via C-terminal twin-strep tags. The individual numerical values and original images for the data presented in this figure may be found in S1 Data and S1 Raw Images, respectively. CFU, colony-forming unit; OC, open-circle; SC, supercoiled; SEC, size exclusion chromatography; SpyCas9, *Streptococcus pyogenes* Cas9. (PDF)

**S11 Fig. Divalent cations influence AcrIIA22's nicking activity.** (**A**) We present the impact of different divalent cations on AcrIIA22's nicking activity, which is highest with $Mg^{2+}$, $Mn^{2+}$, and $Co^{2+}$. (**B**) The OC plasmid product persists through phenol-chloroform extraction following AcrIIA22 treatment, indicating that it directly results from AcrIIA22's nicking activity. OC, open-circle; SC, supercoiled. (PDF)

**S1 Table. Whether known anti-CRISPRs can bind Cas proteins or inhibit their cleavage activity as purified proteins.** (PDF)

**S2 Table. PC4-like proteins with structural homology to AcrIIA22.** (PDF)

**S3 Table. Descriptions of all sequences used in this study.** All sequences and annotations are also available as supplemental data. (PDF)

**S4 Table. Plasmids used in this study.**
(PDF)

**S5 Table. Gene sequences used in this study.**
(PDF)

**S1 Data. All raw data for main and supplemental figures depicted in this study (as a spreadsheet).**
(XLSX)

**S2 Data. 68 contigs sequences referenced in the manuscript with Pfam, TIGRFAM, and AcrIIA22 homolog annotations (in genbank format).**
(GB)

**S3 Data. 68 contigs sequences referenced in the manuscript (in fasta format).**
(FASTA)

**S4 Data. Nine AcrIIA22-encoding prophage sequences referenced in the manuscript with Pfam, TIGRFAM, and AcrIIA22 homolog annotations (in genbank format).**
(GB)

**S5 Data. Nine AcrIIA22-encoding prophage sequences referenced in the manuscript (in fasta format).**
(FASTA)

**S6 Data. All metagenomic contigs with ≥98% nucleotide identity to *acrIIA22*-associated genes, *purF* and *radC*. Pfam, TIGRFAM, and AcrIIA22 homolog annotations are also provided (file in genbank format).**
(GB)

**S7 Data. Amino acid sequence alignment of 30 AcrIIA22 homologs (in fasta format).**
(FASTA)

**S8 Data. The detailed PDB validation report for AcrIIA22's crystal structure.**
(PDF)

**S1 Raw Images. Full gel images for all cropped gels depicted in this study, compiled into a .pdf document.**
(PDF)

## Acknowledgments

We thank Kaylee Dillard, Ilya Finkelstein, Yu-Ying Phoebe Hsieh, Tera Levin, and Janet Young for comments on the manuscript.

## Author Contributions

**Conceptualization:** Kevin J. Forsberg, Brett K. Kaiser.

**Data curation:** Kevin J. Forsberg, Danica T. Schmidtke, Rachel Werther, Barry L. Stoddard, Brett K. Kaiser.

**Formal analysis:** Kevin J. Forsberg, Danica T. Schmidtke, Rachel Werther, Brett K. Kaiser.

**Funding acquisition:** Barry L. Stoddard, Harmit S. Malik.

**Investigation:** Kevin J. Forsberg, Danica T. Schmidtke, Rachel Werther, Ruben V. Uribe, Deanna Hausman, Barry L. Stoddard, Brett K. Kaiser.

**Methodology:** Kevin J. Forsberg, Danica T. Schmidtke, Rachel Werther, Deanna Hausman, Brett K. Kaiser.

**Project administration:** Kevin J. Forsberg, Barry L. Stoddard, Harmit S. Malik.

**Resources:** Kevin J. Forsberg, Barry L. Stoddard, Harmit S. Malik.

**Software:** Kevin J. Forsberg.

**Supervision:** Kevin J. Forsberg, Morten O. A. Sommer, Barry L. Stoddard, Harmit S. Malik.

**Validation:** Kevin J. Forsberg, Brett K. Kaiser.

**Visualization:** Kevin J. Forsberg, Danica T. Schmidtke, Rachel Werther, Barry L. Stoddard, Brett K. Kaiser, Harmit S. Malik.

**Writing – original draft:** Kevin J. Forsberg.

**Writing – review & editing:** Kevin J. Forsberg, Danica T. Schmidtke, Rachel Werther, Deanna Hausman, Barry L. Stoddard, Brett K. Kaiser, Harmit S. Malik.

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
