## [Editor Report · Decision Letter 0]

26 Jan 2021

Dear Dr Forsberg, 

Thank you for submitting your manuscript entitled "AcrIIA22 is a novel anti-CRISPR that impairs SpyCas9 activity by relieving DNA torsion of target plasmids" for consideration as a Research Article by PLOS Biology.

Your manuscript has now been evaluated by the PLOS Biology editorial staff as well as by an academic editor with relevant expertise and I am writing to let you know that we would like to pursue your manuscript further and invite you to submit a revised version of the manuscript.

However, before we can invite a revision, we need you to complete your submission by providing the metadata that is required for full assessment. To this end, please login to Editorial Manager where you will find the paper in the 'Submissions Needing Revisions' folder on your homepage. Please click 'Revise Submission' from the Action Links and complete all additional questions in the submission questionnaire.

Please re-submit your manuscript within two working days, i.e. by Nov 27 2020 11:59PM.

Kind regards,

Richard Hodge, PhD

Associate Editor

PLOS Biology

---

## [Editor Report · Decision Letter 1]

29 Jan 2021

Dear Dr Forsberg,

Thank you very much for submitting your manuscript "AcrIIA22 is a novel anti-CRISPR that impairs SpyCas9 activity by relieving DNA torsion of target plasmids" for consideration as a Research Article at PLOS Biology.

As you know, your manuscript and plan of revision have been evaluated by the PLOS Biology editors and by an Academic Editor with relevant expertise. I apologise for the time it has taken us to return you with a decision, which was due to in-depth discussions we had with our Academic Editor about the necessary revisions.

Based on your responses to the reviews from Reviews Commons, we would be happy to invite revision of your study along the lines indicated by the reviewers. In addition, the Academic Editor has provided detailed comments on the manuscript, which are pasted below alongside the Review Commons reviews and should also be addressed. Please note that we cannot make a decision about publication until we have seen the revised manuscript; your revised manuscript is also likely to be sent for further evaluation by the original reviewers.

While we are editorially quite interested in the work, revision of the manuscript would need to provide further evidence for the proposed nickase activity of AcrIIA22, as emphasized by the Academic Editor and the reviewer reports, which goes along the lines of your revision plan. The Academic Editor has also raised concerns about likelihood that AcrIIA22 functions as an anti-CRISPR in a physiological setting, and this would either need to be experimentally addressed or discussed in detail and caveated in the manuscript text. Depending on the scope and outcome of the revisions, the title may need to change to better reflect the data provided, as the Academic Editor considers that some claims (e.g. the anti-CRISPR function of the protein) are not sufficiently supported.

Whilst we have provided a time frame of 3 months to receive your revised manuscript, we are certainly open to offering an extension to this deadline should you require more time to complete your revisions. Please email us (plosbiology@plos.org) if you have any questions or concerns, or would like to request an extension. At this stage, your manuscript remains formally under active consideration at our journal; please notify us by email if you do not intend to submit a revision so that we may end consideration of the manuscript at PLOS Biology.

*Re-submission Checklist*

*Published Peer Review*

*PLOS Data Policy*

*Blot and Gel Data Policy*

Sincerely,

Richard

Richard Hodge, PhD

Associate Editor

PLOS Biology

Comments from the Academic Editor 

1. The authors have not proved that AcrIIA22 is a nickase. The proteins that are close in structure are not nickases. There is no obvious active site in AcrIIA22. The authors do no not discuss sequence similarity between their anti-CRISPR and any enzyme that cuts DNA. I assume that there is none. The biochemistry is not convincing. To support that AcrIIA22 is a nickase, careful fractionionation of preparations through at least two columns would need to be documented. The peak of nickase activity needs to correlate with the peak of AcrIIA22 protein elution. Nickase activity needs to be assayed in all fractions especially those with little or no AcrIIA22 in them. Nickase could easily come from a contaminating protein present at low abundance. Assuming that AcrIIA22 is a nickase, what is the level of activity (e.g. rate of nick introduction/weight of protein/time). How does this rate compare to known nickases. Purification of the deletion mutant of AcrIIA22 (2 aa truncation) is not a valid control. This protein has a different oligomerization state. Proteins co-purifying with it may be completely different than wild-type. The protein could also be partially unfolded or much less stable than wild-type, which would also alter its chromatographic behaviour. For a valid comparison of purified proteins, the authors need a mutant (preferably a single amino acid substitution) that abrogates nickase activity but has no effect on oligomerization or protein stability. Protein stability needs to be accurately assessed using biophysical methods. Since the authors are proposing a very unusual anti-CRISPR mechanisms that is only weakly supported by the other data presented, their biochemistry must be absolutely rigorous. The authors should also determine whether AcrIIA22 is co-purifying with nucleic acid. Removal of nucleic acid during purification may be important as other DNA-interacting proteins may be co-purifying with bound nucleic acid. Does AcrIIA22 bind to DNA?

2. The proposed mechanism of AcrIIA22 function is tenuous. SpyCas9 cuts linear DNA well in vitro as the authors show. The cutting of open circles by SpyCas9 in vitro is not reduced under all conditions. I find it hard to imagine how a presumably non-specific nickase would be able to provide just the right amount of nicking in vivo to inhibit SpyCas9 but not inhibit other processes that are crucial for the phage or plasmid. To convincingly argue about the mechanism of AcrIIA22 , the authors would need some bona fide in vivo system where they would be able to correlate nick formation with anti-CRISPR activity. They have not done this even in the artificial in vivo assay in E. coli, and so this -if not experimentally proven- would need to be appropriately discussed and caveated.

3. The results do not convincingly support that the natural function of AcrIIA22 is as an anti-CRISPR. This protein is likely involved in recombination functioning as a single-stranded DNA-binding protein as suggested by its structure. Most phages encode recombination proteins. When over-expressed in E. coli, AcrIIA22 does inhibit SpyCas9, but it may inhibit many other DNA-interacting enzymes. It could be binding all over the plasmid DNA and inhibiting SpyCas9 that way. Given that AcrIIA22 homologs are mostly encoded in phages, it is discouraging that it does not inhibit phage replication particularly well even when overexpressed from a plasmid. To function within the context of a phage infection, high potency is required as the anti-CRISPR is initially present at very low concentration after genome injection. How do the authors consider that this may be working? Ideally, data would be provided to support a role during infection, but at a minimum they would need to discuss AcrIIA22 function in this context.

Reviewer Comments from Review Commons

Reviewer #1

Forsberg et al. describe the characterization of a small 54-aa protein, encoded by hypervariable regions of clostridial bacteria and their prophages, that was selected from a metagenome library based on its capacity to completely protect a plasmid from being degraded by SpCas9. The authors call the protein an anti-CRISPR protein, AcrIIA22. However, characterization of the protein reveals a rather general mode of action. It does not, like many Acr proteins, directly interact with some of the key proteins of the CRISPRCas machinery. Its crystal structure reveals homology to PC4-like nucleic acid binding proteins such as the single-stranded binding protein of E.coli phage T5 that is involved in unwinding of dsDNA, and/or recombination. Analysis of the AcrIIA22 protein reveals it is has nickase activity that result in relaxation of supercoiled dsDNA. This activity appears to cause a substantial decrease in interference by SpCas9, protecting the plasmid from being cleaved. At the same time, it also caused slight toxicity as deduced from a 7% drop in growth rate. This is an interesting study, well executed and presented. There are some matters that should be addressed as indicated below.

**Major comments**

1) As the authors admit, there is no direct interaction of AcrIIA22 with the type IIA Cas9 nuclease. Hence, the question is if this protein should be called an AcrIIA system, or even whether it should be classified as an anti-CRISPR protein. Given the indirect effect on Cas9 targeting, it is anticipated that the nickase activity will affect all enzymes that rely on a negatively supercoiled dsDNA configuration, possibly including non-CRISPR defence systems, but certainly including Type IE-IF Cascade/Cas3 systems (REF 32), Type IIA-C Cas9 systems (REFS 13-16). In addition, one would assume that also Type V Cas12 systems would be sensitive to the topology of their dsDNA target. So at best the name should be "general Acr protein". 

2) To make claims that the AcrIIA22 protein is more potent to protect MGEs against Type II Cas9 than against Type I Cascade, either this experiment should be done, or the claims (line 343-345) should be toned down. 

3) A crystal structure of AcrIIA22 is provided, evidence is provided for metal-dependent nickase activity, and alignments have been made (to build a phylogenetic tree) of closely and distantly related sequences. Still, nothing is mentioned about conserved (metalbinding) residues that might constitute the catalytic site that allows for nicking. At least an in silico analysis should be included, and whenever possible an experimental verification of the prediction. 

4) Line 110 - Details should be provided on the genetic material of phage Mu (RNA/ssDNA/dsDNA, linear/circular), as basis for discussion why AcrIIA22 does not seem to protect it from Cas9 cleavage. **Minor comments** Line 55 - provide detail on 'novel mechanism' of AcrIIA11 Line 72 - what inducible promoter? 

Significance (Required): This is an interesting study, well executed and presented. There are some matters that should be addressed as indicated above. My expertise is Mol Microbiology, with focus on CRISPR-Cas biochemistry.

Reviewer #2

Anti-CRISPRs (Acrs) are small proteins harbored by phages that inactivate CRISPR-Cas systems and thereby enable tolerance of infection. Identifying Acrs and understanding their mechanism of action has proven to be important for both understanding host/phage interactions as well as controlling CRISPR-based biotechnologies, most notably, Cas9- mediated genome editing. The known Acrs are diverse in their amino acid sequence, protein structure, and mechanism of Cas inhibition. The authors recently described a genetic approach which identified a new Acr (AcrIIA11) by screening metagenomic libraries for inhibition of Cas9-mediated plasmid cleavage in E. coli (Forsberg et al. Elife 2019). 

In this manuscript, the authors characterized a second metagenomic contig that was hit in the screen. They identified a single gene (AcrIIA22) from this contig which is responsible for inhibiting Cas9-mediated interference of the targeted plasmid and to a lesser extent, Mu phage. Using thorough bioinformatic analysis the authors detected several functional AcrIIA22 homologs in prophages and hypervariable islands within the Clostridial genus CAG-217. The authors purified the AcrIIA22 protein and determined its crystal structure, found that it forms a tetramer in solution, and identify residues required for oligomerization. Unlike most Acrs, this one did not exhibit detectable binding to its cognate Cas protein and did not inhibit Cas9 cleavage of linear DNA substrates. Instead, the authors used a structural homology search and plasmid mobility assays (both in vivo and in vitro) to show that AcrIIA22 possesses DNA nickase activity. They perform experiments on prenicked plasmid substrates that suggest AcrIIA22 inhibits Cas9 by removing the DNA supercoils that Cas9 requires for maximal activity. 

**Major comments:** 

The in vivo experiments convincingly establish that AcrIIA22 is a member of a family of anti-CRISPR proteins that inhibit Cas9 in both anti-plasmid and anti-phage interference. The in vitro data demonstrating that AcrIIA22 is a nickase are compelling as well. The ∆2aa mutant data suggest that oligomerization is required for both nicking and Cas9 inhibition. I commend the authors on their thorough exploration of AcrIIA22 function and evolution. 

I have some reservations regarding the claim that AcrIIA22-nicked plasmids are resistant to Cas9 cleavage. The major piece of experimental evidence supporting this idea is presented in Figure 6C, where plasmid substrates were pre-nicked by either AcrIIA22 or another nickase, and then are tested as substrates in SpyCas9 DNA cleavage assays. The manuscript text states "SpyCas9 showed a clear preference for cleaving the supercoiled substrate versus the AcrIIA22-treated open-circle plasmid", but after looking at the figure (either on screen or a print-out) I had some difficulty agreeing with this assessment. This is largely because in the reactions presented, Cas9-mediated cleavage of the un-treated substrate (Fig 6C - lane 5) has not gone to completion: most of the supercoiled substrate remains uncleaved and there is a fairly weak linear cleavage product. Both of the pre-nicked substrates (lanes 4 and 6) also appear to have weak linear product bands. To my eye, there is very slightly more linear Cas9 cleavage product for the supercoiled substrate than there is for pre-nicked. The authors note in the Figure 6 legend that some of the Cas9 cleavage product is open-circular (a single strand cut). However, the untreated plasmid (lane 2) also contains open-circular DNA. Altogether it is difficult to tell whether the nicked plasmid is indeed any less sensitive to Cas9 than the supercoiled form. A few suggestions / points related to this: 

1.Can the reactions be run for longer time periods to generate more linear Cas9 product? This should make the difference between the nicked and supercoiled substrates more apparent, if the nicked substrates are indeed resistant to cleavage. 

2.The experiment shown in Fig S10B (same experiment as Fig 6C, different reaction conditions) seems to suggest that pre-nicked substrates are still susceptible to cleavage by Cas9. The authors state in the legend that "the reaction proceeded too quickly to detect SpyCas9's substrate preference", however the reaction is only at about 50% completion and the three substrates have been cleaved equally. Why shouldn't the interpretation be that Cas9 prefers neither substrate? 

3.The authors state that AcrIIA22 "capitalizes on SpyCas9's stringent requirement for negative supercoils to form a productive R-loop". Does SpyCas9 strictly require negative supercoiling for R-loop formation and DNA cleavage? SpyCas9 is capable of cleaving short linear DNA substrates, which should have minimal supercoiling. Previous singlemolecule studies investigating Cas9's supercoiling dependence (references 13-16) seem to suggest that rather than a stringent topology requirement, Cas9 binding and cleavage is enhanced by negative supercoiling. Thus, perhaps a quantitative assessment of cleavage kinetics for each substrate would be revealing? 

Significance (Required): Anti-CRISPRs hold promise for enhanced control of Cas9-mediated genome editing. The discovery of a new Acr with a new mechanism of action could lead to tighter off-switches for Cas9, especially when used in combination with other known Acrs. Understanding the effects of DNA topology on Cas9 function will inform future technological uses of Cas9. 

**Referees cross-commenting** I agree with Reviewer #1's major comments 1-2 regarding the expected general antinuclease property of AcrIIA22. Testing the inhibitory activity of AcrIIA22 against other CRISPR-Cas types would improve the study

Reviewer #3

**Summary:** 

Forsberg et al. describe a new anti-CRISPR protein AcrIIA22, which was identified in a previously reported screen for anti-CRISPR effectors. A genomics analysis revealed that the presence of AcrIIA22 is a feature shared by diverse hypervariable regions of clostridial genomes. Crystal structure of AcrIIA22 is presented with the main finding being the structural basis of its oligomerization. This information was used to design a deletion mutant which did not oligomerize and was unable to inhibit Cas9. The authors also show that in vitro plasmids pre-treated with AcrIIA22 are nicked and less susceptible to Cas9 cleavage. These results lead to a model in which AcrIIA22 nicks DNA to relieve its supercoiling required for Cas9 activity. This inhibits Cas9 activity which would constitute a novel mode of indirect anti-CRISPR action. 

**Major comments:**

The main concern about this work is the proposed nickase activity of AcrIIA22. PC4 proteins are known to bind nucleic acids but have not been shown to have enzymatic activities. The basis of the proposed enzymatic activity of AcrIIA22 is in fact not even discussed in the paper. The metal dependence of the nickase activity (Suppl. Fig 9) follows a pattern typical for many metal-dependent nucleases with Mg2+ supporting the activity and Ca2+ being inhibitory. This would imply that AcrIIA22 is a typical nuclease with a metal ion-binding active site. No such active site has been identified in PC4-related proteins and it is not described in the current work based on the AcrIIA22 structure itself. To support the main conclusions of the manuscript, the authors need to better document AcrIIA22 nuclease activity for example by relatively simple experiments with short synthetic DNA substrates (single- and double-stranded). It is also very important to exclude the possibility that the nickase activity of purified AcrIIA22 comes from impurities in the protein preparation. DNase is a common activity in living cells. As a minimum, the quality of the protein needs to be shown in the Supplementary Information. The possible basis of the proposed enzymatic activity of PC4 needs to be thoroughly discussed, preferably based on the structural data.

A key element of the proposed mechanism is lower activity of Cas9 on open circle form of plasmid DNA. It is concerning that under certain conditions (Supp. Fig 10B), supercoiled and and nicked plasmids are cleaved by Cas9 equally well. The authors argue that this was due to the high reaction rate. This needs to be confirmed by time-course experiments at lower temperature which should slow down the reaction. 

Multiple structures of PC4 proteins in dimeric state and with nucleic acid bound are available (for example PMID: 16415882). The dimer architecture is conserved between PC4 and PC4-like protein from phage T5, so it is likely a universal feature of these proteins. Figure 4 needs to show more clearly how the typical PC4 dimerization is similar and different from the dimer present in the asymmetric unit of AcrIIA22 structure (Fig. 4A) and the proposed oligomeric state of the protein (Fig. 4D). Would the delta2aa mutant disrupt this canonical dimerization? Is the reported PC4 DNA binding mode compatible with the oligomeric state of AcrIIA22 or do authors imply a novel mode of the interactions with nucleic acid (dashed arrow in Fig 4E)? It is also important to show the ability of AcrIIA22 and its deletion mutant to bind single-stranded DNA. This can be done by a relatively simple extension of the gel filtration experiment shown in Fig 4C with proteinDNA complexes. Preferably, if possible, these experiments should be done with an in-line MALS-based measurement of the molecular weight. This will provide much more accurate MWs determination compared with column calibration traces. 

Samples of electron density maps are not shown in the manuscript and the detailed PDB validation report was not attached to this submission. However, based on Table 1 the structural work appears very solid with good dataset and structure statistics. 

For in vitro experiments the numbers of replicates are not reported and quantification graphs do now show error bars. Quantification of nickase activity of delta2aa variant is shown in Fig 5C. This is an important experiment which may support the notion that the reported activity does indeed come from AcrIIA22 and not from impurities in its preparation. Therefore, the gels used to generate these data need to be shown in Supplementary Information, similar to the experiments for which the quantification is shown in Fig 5C. 

**Minor comments:** 

Authors state the homology to PC4 suggested that AcrIIA22 is a nickase. However, the rationale for this is not clearly described. Otherwise, the presentation of the data is clear and previous work is properly referenced. 

The nature of the proposed mechanism and the results with Mu phage (weak protection by AcrIIA22) suggest that the presence of AcrIIA22 would create an advantage for circular dsDNA phages and only when their genome is present in supercoiled form. The authors may want to discuss this and the prevalence of circular dsDNA phages of Clostridium bacteria. 

Significance (Required): The main finding from the structural part is the basis of AcrIIA22 oligomerization, but it needs to be further analyzed as described above. The fact that AcrIIA22 is similar to PC4- like proteins was likely apparent from the protein sequence because a homology model of the protein could be built which was successfully used for solving the crystal structure by molecular replacement. 

Overall, this is a potentially very interesting work describing a novel mode of indirect CRISPR inhibition by relieving of DNA supercoiling. This could be an important conceptual advance. CRISPR mechanisms are obviously a very timely and popular subject, so this work would be interesting for a wide audience interested in these mechanisms.

Reviewer's expertise: nucleic acid enzymes, protein biochemistry, structural biology. I cannot evaluate the genomics part of the manuscript very thoroughly.

---

## [Decision Letter · Decision Letter 2]

5 Aug 2021

Dear Kevin,

Thank you for submitting your revised Research Article entitled "AcrIIA22 is a novel anti-CRISPR that impairs SpyCas9 activity by relieving DNA torsion of target plasmids" for consideration as a Research Article in PLOS Biology. Please accept my apologies for the delays you have experienced in the processing of your revised manuscript. Your revised study was evaluated by the PLOS Biology editors, an Academic Editor with relevant expertise and by the three reviewers that assessed your work at Review Commons. 

As you will see in their reports at the end of this email, the reviewers are generally satisfied with your responses to their comments and feel that the additional data supports the nickase activity of AcrIIA22. Nevertheless, you will also see from the outstanding concerns of Reviewer 1 and of the Academic Editor (whose comments you will also find at the end of this email) that they remain sceptical as to the native activity of AcrIIA22 and whether it functions as an anti-CRISPR protein in vivo via the proposed mechanism of relieving torsional stress in the DNA. Although we are editorially interested in the work and there is enough reviewer support to move forward, we ask that you carefully consider the concerns raised by the Academic Editor and by Reviewer #1 and respond to the points raised both in your 'Response to Reviewers' file and in the manuscript text by clearly discussing the caveats, recasting and tempering the claims if/as needed. It is ultimately in your best interest to put forth claims that have the best possible chance of standing the test of time. 

In addition, we ask that you please address the following editorial and other policy-related requests that I have provided below:

A) You may be aware of the PLOS Data Policy, which requires that all data be made available without restriction: http://journals.plos.org/plosbiology/s/data-availability. For more information, please also see this editorial: http://dx.doi.org/10.1371/journal.pbio.1001797

Regardless of the method selected, please ensure that you provide the individual numerical values that underlie the summary data for the following Figures, as they are essential for readers to assess your analysis and to reproduce it:

Figure 1B-C, 3B-C, 6A, 6C, 7D-E, Fig S1, S2B, S4C, S9A-B, S10C-D, S10F

B) Please also ensure that each of the relevant figure legends in your manuscript include information on *WHERE THE UNDERLYING DATA CAN BE FOUND*, and ensure your supplemental data file/s has a legend

C) Please ensure that your Data Statement in the submission system accurately describes where your data can be found and is in final format, as it will be published as written there. Please ensure that the structural data that is currently deposited at the PDB (accession number 7JTA) is made publicly available at this stage, as it is currently on hold.

D) We require the original, uncropped and minimally adjusted images supporting all blot and gel results reported in the following Figures:

Figure 5A-C, 7A, 7C, Fig S2C-D, S6, S7B, S7D, S10A-B, S10E, S10G-H

We will require these files before a manuscript can be accepted so please prepare and upload them now. Please carefully read our guidelines for how to prepare and upload this data: https://journals.plos.org/plosbiology/s/figures#loc-blot-and-gel-reporting-requirements

------

We expect to receive your revised manuscript within one month, but please let us know if you would need more time, especially given the circumstances of your relocation.

*Published Peer Review History*

*Early Version*

Sincerely,

Richard

Richard Hodge, PhD

Associate Editor, PLOS Biology

rhodge@plos.org

Reviewer remarks:

Reviewer #1 (John Van der Oost, signs his review): I am not satisfied with the authors' response to some of my earlier comments, again copied below:

**Major comments**

4) As the authors admit, there is no direct interaction of AcrIIA22 with the type IIA Cas9 nuclease. Hence, the

question is if this protein should be called an AcrIIA system, or even whether it should be classified as an

anti-CRISPR protein. Given the indirect effect on Cas9 targeting, it is anticipated that the nickase activity will

affect all enzymes that rely on a negatively supercoiled dsDNA configuration, possibly including non-CRISPR

defence systems, but certainly including Type IE-IF Cascade/Cas3 systems (REF 32), Type IIA-C Cas9

systems (REFS 13-16). In addition, one would assume that also Type V Cas12 systems would be sensitive to

the topology of their dsDNA target. So at best the name should be "general Acr protein".

AUTHOR RESPONSE: We have followed the reviewer's suggestion and have tested AcrIIA22 against additional CRISPR-Cas systems

(revised Figure 3C). We chose type II and type V systems, as these are the only ones naturally occurring with its

host CAG-217 genus. We found that AcrIIA22 can protect plasmids from Cas9, but not Cas12-based systems.

Thus, we have elected to keep the type II nomenclature for AcrIIA22.

REVIEWER RESPONSE: The data provided on Cas12 are convincing, their activity seems not to be affected by the nickase. Still, the currently used name (AcrIIA22) suggests that the protein specifically counteracts Type IIA systems. However, it also counteracts the NmCas9, which is Type IIC. Hence, it seems to be a rather generic nickase that affects the functionality of enzymes that rely on negative supercoiling (including Type I, see below).

5) To make claims that the AcrIIA22 protein is more potent to protect MGEs against Type II Cas9 than

against Type I Cascade, either this experiment should be done, or the claims (line 343-345) should be toned

down.

AUTHOR RESPONSE: We thank the reviewer for their suggestion. We have amended our language to make this point clearer. We

now refer to published in vitro data that demonstrates Cas9 is more susceptible to supercoil state than type I

Cascade. 

REVIEWER RESPONSE: It is very difficult to compare experiments that have been independently reported in the literature. As said before, either experiments have to be included in which a direct comparison of Cas9 and Cascade is performed, or the clains that this nickase only affects Cas9 should be rephrased in: "this nickase affects the functionality of proteins/enzymes that rely on negative supercoiling, including Cas9, but probably also other (Cas/non-Cas proteins/enzymes".

Reviewer #2: The authors have gone to great lengths experimentally to address each of the points raised in review. The new data has convinced me that AcrIIA22-nicked plasmids are less sensitive to Cas9 cleavage in vitro. The additional mutants and biochemical analyses are compelling evidence of AcrIIA22-mediated nicking, and in every case, there is a strong correlation between nickase activity and Cas9 inhibition. The demonstration that AcrIIA22 is functional in preventing targeting by NmeCas9 and SpyCas9, but not either Cas12 homolog, argues that AcrIIA22 is not simply coating DNA and non-specifically interfering with nuclease activity. Overall, the authors have provided a thorough mechanistic characterization of AcrIIA22 and provided solid evidence to support their claims; therefore I recommend publication.

Reviewer #3: The authors addressed my comments appropriately. In particular they provided additional experiments supporting the nickase activity and prepared panels with structural analysis.

Academic Editor Comments: This revised manuscript is improved over the original. The authors have provided much stronger evidence that AcrIIA22 is a nickase. They also show a convincing correlation between the nickase activity of AcrIIA22 and its plasmid-rescuing activity in vivo.

Despite the improvements in the manuscript, I am still entirely unconvinced by the proposed mechanism for this Acr. In addition, there is also no reason to believe that this protein functions as an anti-CRISPR within the context of the phages where its gene is located. These are the problems:

1. AcrIIA22 displays absolutely no in vitro inhibition of SpyCas9 activity on linear dsDNA.

2. AcrIIA22 only very weakly prevents SpyCas9 from blocking phage infection.

3. The in vitro assay for AcrIIA22 activity on super-coiled DNA does not seem physiologically relevant. As I understand it, the authors first treat DNA with AcrIIA22, then purify the open circle DNA produced. This DNA is then treated with SpyCas9. In vivo, whether a plasmid is introduced into a cell by conjugation, or a phage genome is introduced by injection, the CRISPR-Cas system will be present before any Acr is produced. Thus, an Acr must have strong inhibitory activity in the presence of the CRISPR-Cas system. Pre-treating DNA with AcrIIA22 and then treating it with SpyCas9 is entirely artificial with respect to the in vivo situation. What would happen if SpyCas9 and AcrIIA22 were added to a reaction at the same time?

4. There is no assay where the authors treat linear DNA, open circle DNA, and supercoiled DNA with SpyCas9 under the same conditions and run on the same gel. What is SpyCas9 most active in cleaving? From what is shown, it looks like SpyCas9 cleaves linear DNA faster than supercoiled. I do not understand how a DNA cleaving enzyme would cleave supercoiled and linear DNA well, but cleave open circles poorly. Topologically, I don’t see how linear DNA and open circle DNA would be distinguishable in vitro. They are both fully relaxed. Or am I missing something? The authors must provide an explanation for this. The authors say, “Our findings are consistent with previous in vitro experiments, which indicated that Cas9 requires a higher degree of negative supercoiling than type I CRISPR-Cas systems to provide the free energy needed for R-loop formation.” If this were true, wouldn’t Cas9 cut supercoiled DNA much better than linear? Why is it only open circle that seems to be cut poorly?

5. The authors state that “we show that AcrIIA22 acts by nicking supercoiled DNA to relieve torsional stress on a target plasmid, thereby impairing SpyCas9 activity in vivo and in vitro.” This is misleading. The authors have not shown that the relieving of torsional stress is inhibiting SpyCas9 activity in vivo. They have shown a correlation between nicking activity in vitro and plasmid rescue in vivo. They have no evidence that SpyCas9 is responding to the degree of torsional stress in vivo. In fact, the torsional state of the plasmids in vivo has not been assayed at all. The only “in vivo” assay of torsional stress comes from plasmid isolation after growth in the presence of AcrIIA22. It is possible that the plasmid nicking observed on gels occurs during the preparation of the DNA (i.e. AcrIIA22 nicks the DNA in vitro after cell lysis). Maybe the nickase activity of AcrIIA22 stimulates plasmid recombination which helps overcome SpyCas9 cleavage.

6. To explain how AcrIIA22 works, the authors write, “We speculate that AcrIIA22 treads a fine balance between activity and toxicity; its nicking activity is high enough to antagonize SpyCas9 in a kinetic race, but not so high that it would be toxic to the host cell (Supplemental Figure 1). To strike this balance, AcrIIA22’s nicking activity may somehow be regulated in vivo, perhaps via transient interactions with Cas9, other host factors, or its oligomerization”. If AcrIIA22 must tread such a fine balance in vivo that might involve host factors, why is it working in a totally artificial system in which it is overexpressed and blocking a type of Cas9 that it would not encounter in nature? I still find it hard to imagine how a presumably non-specific nickase would be able to provide just the right amount of nicking in vivo to inhibit Cas9 but not inhibit other processes that are crucial for the phage or plasmid. To me, this mechanism appears to be fundamentally infeasible.

To conclude, I do not doubt that AcrIIA22 inhibits SpyCas9-mediated plasmid loss in E. coli and I accept that AcrIIA22 has nickase activity in vitro. I am not convinced that relief of torsional stress is the mechanism for the in vivo activity of AcrIIA22 or that the biological role of AcrIIA22 is to act as an anti-CRISPR. Given that the title of this paper is “AcrIIA22 is a novel anti-CRISPR that impairs SpyCas9 activity by relieving DNA torsion of target plasmids”, my objections create a problem for publication of the manuscript in its current state.

---

## [Editor Report · Decision Letter 3]

28 Sep 2021

Dear Kevin,

Thank you very much for your patience as we searched for an alternative Academic Editor to handle your submission, after the previous Guest Academic Editor unexpectedly had to step down. I hope that this decision reaches you in time for your upcoming award deadline. 

On behalf of my colleagues and the new Academic Editor, Alexander Meeske, I am pleased to say that we can in principle offer to publish your Research Article "The novel anti-CRISPR AcrIIA22 relieves DNA torsion in target plasmids and impairs SpyCas9 activity" in PLOS Biology, provided you address any remaining formatting and reporting issues. These will be detailed in an email that will follow this letter and that you will usually receive within 2-3 business days, during which time no action is required from you. Please note that we will not be able to formally accept your manuscript and schedule it for publication until you have made the required changes.

I have also pasted some comments from Reviewer #1 below my signature - as you will see, they remain sceptical of the claim that AcrIIA22 naturally functions as an anti-CRISPR protein. However, the new Academic Editor feels the text is sufficiently caveated and contextualized for publication and we have decided to proceed with the acceptance of your manuscript. We decided to share this opinion with you for the sake of transparency, in case it is useful in anticipating the reaction of some of the research community to your paper.

PRESS

Sincerely, 

Richard

Richard Hodge, PhD 

Associate Editor, PLOS Biology

rhodge@plos.org

PLOS

Reviewer #1 Comments: 

I stick to my conclusion that the Authors cannot call this protein an anti-CRISPR protein - it is just a nickase, that indirectly affects the functionality of Cas9 nucleases, and most likely other DNA-binding proteins that require negative supercoiling of the DNA.